# Microbial-driven preterm labour involves crosstalk between the innate and adaptive immune response

Denise Chan[1], Phillip R. Bennett [1,2], Yun S. Lee[1,2], Samit Kundu[1,2], T. G. Teoh[1,2,3], Malko Adan[1,2], Saqa Ahmed[1], Richard G. Brown[1], Anna L. David [4], Holly V. Lewis[1], Belen Gimeno-Molina[1,2], Jane E. Norman[5,6], Sarah J. Stock [5], Vasso Terzidou [1,2,7], Pascale Kropf[2,8], Marina Botto [2,9], David A. MacIntyre [1,2] & Lynne Sykes [1,2,3✉]

There has been a surge in studies implicating a role of vaginal microbiota in spontaneous preterm birth (sPTB), but most are associative without mechanistic insight. Here we show a comprehensive approach to understand the causative factors of preterm birth, based on the integration of longitudinal vaginal microbiota and cervicovaginal fluid (CVF) immunophenotype data collected from 133 women at high-risk of sPTB. We show that vaginal depletion of *Lactobacillus* species and high bacterial diversity leads to increased mannose binding lectin (MBL), IgM, IgG, C3b, C5, IL-8, IL-6 and IL-1β and to increased risk of sPTB. Cervical shortening, which often precedes preterm birth, is associated with *Lactobacillus iners* and elevated levels of IgM, C3b, C5, C5a and IL-6. These data demonstrate a role for the complement system in microbial-driven sPTB and provide a scientific rationale for the development of live biotherapeutics and complement therapeutics to prevent sPTB.

---

[1] Imperial College Parturition Research Group, Institute of Reproductive and Developmental Biology, Department of Metabolism, Digestion and Reproduction, Imperial College London, London, UK. [2] March of Dimes Prematurity Research Center at Imperial College London, London, UK. [3] St. Mary's Hospital, Imperial College Healthcare NHS Trust, London, UK. [4] Elizabeth Garrett Anderson Institute for Women's Health, University College London, London, UK. [5] University of Edinburgh Usher Institute, Edinburgh, UK. [6] Faculty of Health Sciences, University of Bristol, Bristol, UK. [7] Chelsea & Westminster Hospital, Imperial College Healthcare NHS Trust, London, UK. [8] Department of Infectious Diseases, Imperial College London, London, UK. [9] Department of Immunology and Inflammation, Imperial College London, London, UK. ✉email: l.sykes@imperial.ac.uk

The global preterm birth rate is estimated at over 10%[1]. Preterm birth is the leading cause of neonatal and childhood mortality[2]. A third of preterm births are medically indicated for maternal or fetal reasons such as preeclampsia or intrauterine growth restriction. The remainder are considered to be spontaneous, with preterm prelabour rupture of fetal membranes (PPROM) preceding 25–30% of cases[3]. Three physiological processes are required for labour: cervical shortening and dilation, uterine contractility and rupture of fetal membranes. In the case of preterm labour, these occur as a result of a pathological process, and each may occur in isolation, conferring a greater risk of preterm birth. Cervical shortening, a prerequisite of cervical dilation, when detected at 24 weeks leads to an almost tenfold increased risk of preterm birth, with ultrasound detection commonly used as a screening tool[4]. PPROM occurs in 3% of all pregnancies, of whom 50–60% will deliver within a week[5]. There have been limited advances in prediction, prevention, and treatment of spontaneous preterm birth (sPTB) over the last few decades. This is predominantly because it may be due to one of, or a combination of, multiple aetiological factors, yet research studies exploring pathophysiology or therapeutics typically fail to attempt to phenotype cases. Increased understanding of aetiology is required to improve patient stratification for targeted therapeutic intervention, and for the development of novel therapeutic strategies.

Decades of research has implicated inflammation and infection in the aetiology of a significant proportion of spontaneous preterm births[6], especially those occurring prior to 34 weeks[7]. The concept of ascending bacterial infection and/or inflammation from the vagina through the cervix and into the uterine cavity is widely accepted and is supported by both animal and human studies[8,9]. Recent studies of the pregnancy vaginal microbiota have implicated *Lactobacillus* dominance and inhibition of pathogen colonisation of the vaginal niche as an important mediator of preterm birth risk. We, and others have identified *Lactobacillus crispatus* as being especially protective against early onset neonatal sepsis associated with PPROM, cervical shortening and sPTB[10–12]. In contrast, *Lactobacillus* depleted, and high diversity vaginal microbiota are associated with an increased risk of PPROM and of sPTB[13–17].

Despite a large body of evidence supporting the role of vaginal microbiota in sPTB, the mechanism for this remains poorly understood. A cluster of studies have reported associations between *Lactobacillus* deplete or dysbiotic vaginal microbiota and local inflammation at the cervicovaginal interface in the context of PTB, however, immunophenotyping has been limited to analysing cervicovaginal cytokines, chemokines and β defensin[12,18,19].

In this study, we show how the maternal host immune system responds to healthy commensals and pathobionts, and how this interaction influences the risk of sPTB. We hypothesise that the complement system facilitates cross-talk between the innate and adaptive immunity in response to vaginal microbiota. We demonstrate that immune activation and a dysregulated immune response involving the lectin mediated pathway, IgM/IgG complex activated classical pathway, and a pro-inflammatory cytokine milieu occurs in microbial-driven sPTB. The wider implications of these results support the potential for therapeutic modulation of local microbial composition and the immune milieu for the prevention of sPTB.

## Results

**Study cohort.** This study was focused upon women undergoing clinical surveillance at preterm birth prevention clinics during their pregnancy because of risk factors associated with a higher than usual risk of sPTB. Principal risk factors included previous sPTB, previous PPROM, previous mid-trimester pregnancy loss (MTL), previous large loop excision of the transformation zone of the cervix (LLETZ), or a combination of these. A total of 133 women, recruited from preterm birth prevention clinics in five UK Hospitals, provided a total of 385 cross sectional sampling points, 126 at timepoint A ($12^{+0}$–$16^{+6}$ weeks), 133 at timepoint B ($20^{+0}$–$24^{+6}$ weeks) and 126 at timepoint C ($30^{+0}$–$34^{+6}$ weeks) (Fig. 1). Longitudinal samples were targeted at timepoints A and B, since women who delivered before 30 weeks would fail to meet timepoint C. Longitudinal samples for timepoint A and B were collected in 122 women. sPTB occurred in 37 women (27.82%) < 37 weeks, 21 (15.79%) < 34 weeks, and 4 (3.01%) < 28 weeks. PPROM occurred in 22 (16.54%) women, and complicated 59.46% of preterm births. Cervical shortening occurred in 53 women, 39.85% of the study cohort, and 59.46% of women who had a sPTB. Of the 96 women who delivered at term, 40 women had intervention with cervical cerclage and/or progesterone (term intervention, TI) and 56 had no intervention (term uncomplicated, TU). Cervical cerclage was placed if there was a clinical history suggestive of an insufficient cervix or if the cervical length was found to be ≤25 mm in length on ultrasound scan. While women who delivered at term without intervention were used as a control group, it is acknowledged that our study cohort was predefined as being at high risk of preterm delivery. There was no significant difference in maternal age ($p = 0.93$), BMI ($p = 0.20$) or ethnicity ($p = 0.09$) between women who delivered spontaneously preterm or at term with or without intervention (Supplementary Table 1). No women were diagnosed with a sexually transmitted disease.

**Local inflammation, vaginal microbial composition, and preterm birth.** Cervicovaginal fluid (CVF) pro-inflammatory cytokine concentrations were compared at each timepoint between women who delivered preterm, at term without intervention and at term with intervention (Supplementary Fig. 1a–d). A significantly higher IL-8, IL-6, and IL-2 was seen at $20^{+0}$–$24^{+6}$ weeks in women who delivered preterm compared to those delivering at term (Supplementary Fig. 1a, b, d). Longitudinal sampling between $12^{+0}$–$16^{+6}$ and $20^{+0}$–$24^{+6}$ weeks revealed a significant increase in IL-8, IL-6, IL-1β and IL-2 between timepoints in women who delivered preterm, and no change in women who delivered at term without intervention (Fig. 2a–d). In contrast, IL-8 and IL-6 were reduced between timepoints A and B in women who received intervention (Fig. 2a, b). No change in cervicovaginal concentrations of IFN-γ, TNF-α, GM-CSF, IL-18, IL-4, and IL-5 were observed (Supplementary Table 2).

Metataxonomic profiling of vaginal bacteria was performed on 385 swabs generating 11,568,580 high-quality reads with an average read count of 30,048 per sample. Using hierarchical clustering of genera-level relative abundance data, samples were grouped as either *Lactobacillus* spp. dominant (>75%), 77% of samples, or *Lactobacillus* spp deplete (≤75%), 23% of samples (Fig. 2e). Both alpha-diversity (Inverse Simpson index) and richness (number of species observed) were significantly higher in *Lactobacillus* deplete compared to *Lactobacillus* dominant vaginal microbiota (Supplementary Fig. 2a, b). At species level, seven community state types (CSTs) were identified, four of which were dominated by a single species of *Lactobacillus*, and three which were not; CST I- *L. crispatus (40%)*, CST II- *L. gasseri (12%)*, CST III -*L. iners (26%)*, CST V -*L. jensenii (6%)*, CST IV-A reflective of high relative abundance of BVAB1 and moderate relative abundance of *G. vaginalis* (0%), CST IV-B reflective of a high relative abundance of *G. vaginalis* and low relative abundance of BVAB1 (9%), and CST IV-C reflective of low relative abundances of *G. vaginalis*, BVAB1 and *Lactobacillus* spp. (6%)), (Fig. 2e).

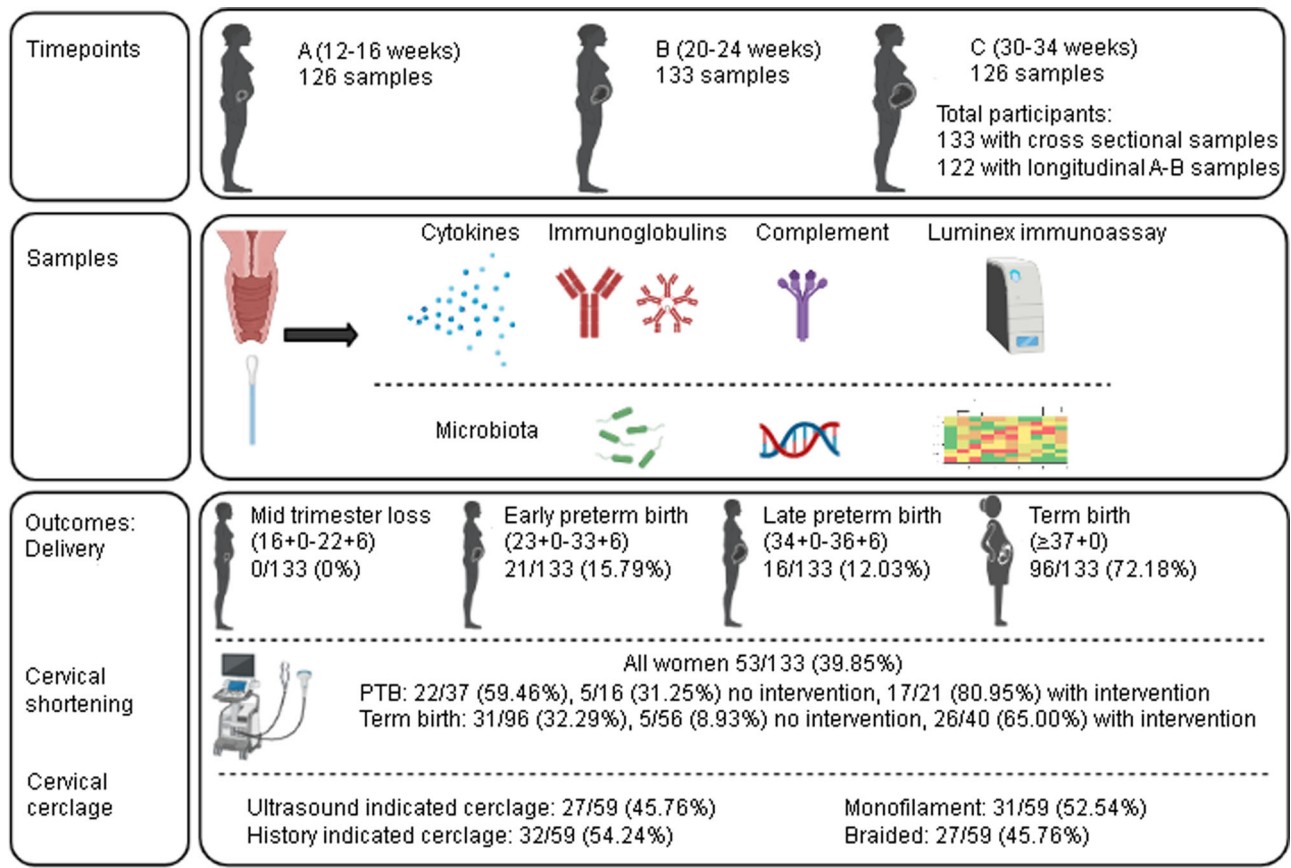

**Fig. 1 Study design.** Women at high risk of preterm birth due to the presence of risk factors were recruited from preterm birth prevention clinics where cervical length ultrasound scans were performed, and cervical cerclage was placed where indicated. Cervicovaginal samples were taken at three timepoints: **A** (12–16 weeks), **B** (20–24 weeks) and **C** (30–34 weeks). Cervicovaginal fluid was analysed for mediators of inflammation (IL-8, IL-6, IL-1β, IL-10, TNF-α, IFN-γ, GM-CSF, IL-5, IL-4, IL-2, IL-18), complement proteins (C3b, C5 and C5a) and mediators of microbial recognition (MBL, IgM, IgG1-4, and IgA). Bacterial DNA was extracted and microbial composition was determined by next generation sequencing. The spontaneous preterm birth rate was 27.82% < 37 weeks and 15.79% < 34 weeks. Of the whole study population, 39.85% of women were identified with cervical shortening (defined as a cervical l length of ≤ 25 mm), compared to 59.46% of women who delivered preterm and 8.93% of women who delivered at term without intervention. Of the cerclages placed, 45.76% were placed due to a short cervix (ultrasound indicated) and 54.24% were placed prior on the basis of clinical history (history indicated). Monofilament was used in 52.54% and braided suture material was used in 45.76% of cases. There was one case where cerclage material type was not known. Figure created with BioRender.com.

CST I was further divided to I-A and I-B sub-CST reflective of degree of *L. Crispatus* dominance, CST III was further divided to III-A and III-B sub-CST reflective of degree of *L.Iners* dominance, and CST IV-C was further divided to IV-C0 IV-C1, IV-C2, IV-C3 and IV-C4 sub-CST reflective of a an even community with *Prevotella* spp., *Streptococcus* spp., *Enterococcus* spp., *Bifidobacterium* spp. or *Staphylococcus* spp., respectively. See Supplementary Data 2 for composition of CSTs and sub-CSTs. Analyses were carried out using the seven community state types, due to none or a small number of samples in sub-CSTs IV -A, CST IV-C0,1,2 and 4. Alpha-diversity of CST I- *L. crispatus* was significantly lower than in every other of the seven CSTs, and richness was lower compared to CST II, IV-B and IV-C (Supplementary Fig. 2c, d).

Twenty-one women delivered before 34 weeks and 16 women delivered between 34 and 37 weeks. There was a greater proportion of women with either CST III or CST IV at $12^{+0}$-$16^{+6}$ (timepoint A) and $20^{+0}$–$24^{+6}$ weeks (timepoint B) in those who had early-preterm birth (Fig. 2f) compared to late preterm birth (Fig. 2g), with a small increase in CST III and CST IV at timepoint B in the former group. However, no significant differences were seen between early-preterm birth and term birth at timepoint A (Fig. 2h). In contrast, women who delivered at term, but who required an intervention had the greatest proportion of CST III/ CST IV at timepoint A (Fig. 2i), which resolved to become comparable with the uncomplicated term group by timepoint B (Fig. 2h). These results suggest that the microbial composition alone is not sufficient to drive sPTB, with the need for other contributing factors such as the differences in host immune responses to play a role.

When all cases were taken together irrespective of outcome, a *Lactobacillus spp.* depleted vaginal composition was associated with significantly higher concentrations of IL-8, IL-6, IL-1β and lower IL-10 (Fig. 3a–d). In women who were *Lactobacillus* deplete, IL-6 and IL-1β were significantly higher by mid gestation in women who delivered preterm compared to those who delivered at term (Fig. 3j, k). Within the CSTs, IL-8 concentrations were significantly higher in CST III and IV-B (but not IV-C), and IL-1β was significantly higher in CST II,III, IV-B and IV-C, regardless of delivery outcome (Fig. 3e–h). In women who had CST III, preterm delivery was associated with higher concentrations of IL-8, IL-6 and IL-1β compared to those who delivered at term (Fig. 3i–k). In women with CST IV IL-8, IL-6 and IL-1β were also significantly higher in those who delivered preterm compared to those who delivered at term (Fig. 3i–k). Sample size limited the ability to compare outcome between sub-CST IV-B and IV-C0-4, however,

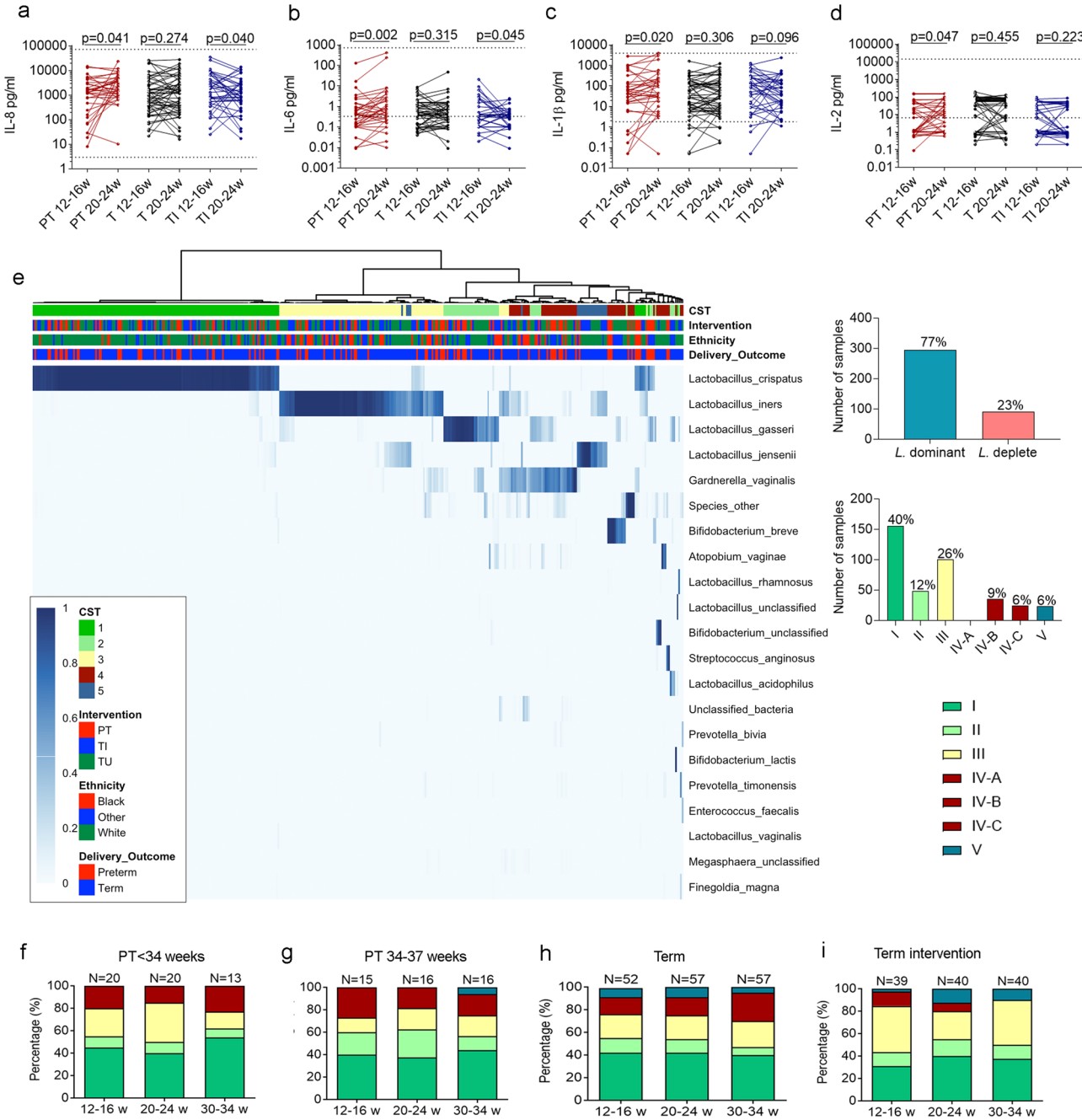

**Fig. 2 Cervicovaginal pro-inflammatory cytokines increase in mid-pregnancy in women who deliver preterm and early-preterm birth is associated with a predominance of *Lactobacillus Iners* or diverse vaginal microbial composition compared to late preterm birth.** Longitudinal sampling of cervicovaginal cytokines was performed in 122 women, of whom 34 delivered preterm, 51 delivered at term without intervention, and 37 delivered at term with an intervention. Concentrations of IL-8 (**a**), IL-6 (**b**), IL-1β (**c**), and IL-2 (**d**) taken at 12–16 weeks and 20–24 weeks are presented. A one-sided Wilcoxon matched pairs signed rank was used for statistical analysis. A heat map shows the distribution of community state types (CST) of studied women, with graphs representing the percentages of samples classed as *Lactobacillus* dominant ($n = 295$) and *Lactobacillus* deplete ($n = 90$), or classed by vaginal microbial composition; CST I (*L. crispatus*) ($n = 155$), CST II (*L. gasseri*) ($n = 48$), CST III (*L. iners*) ($n = 100$), CST IV (diverse) subdivided into CST IV-A ($n = 0$), CST IV-B ($n = 35$), CST IV-C ($n = 24$), and CST V (*L. jensenii*) ($n = 23$), *a total of* $= 385$ samples. **e** Vaginal microbial composition at each timepoint is presented in women who delivered <34 weeks (**f**), 34–37 weeks (**g**), at term without intervention (**h**), and in women who delivered at term following intervention (**i**), $n = 133$ women with cross sectional data. PT = preterm, T = term without intervention, TI = term with intervention. Source data are provided as a Source Data file.

4 of the 6 women who delivered preterm with CST IV had high relative abundance of *G. vaginalis*. compared to only 3 of the 9 women who delivered at term (Supplementary Data 2). No statistically significant differences were seen in the concentration of IL-10 (Fig. 3l). These data demonstrate that a degree of microbial-driven inflammation occurs regardless of outcome, yet

preterm delivery is associated with a significantly greater degree of inflammation, alluding to a dysregulated immune response.

**Mediators of microbial recognition, vaginal microbial composition, and preterm birth.** Concentrations of mannose binding

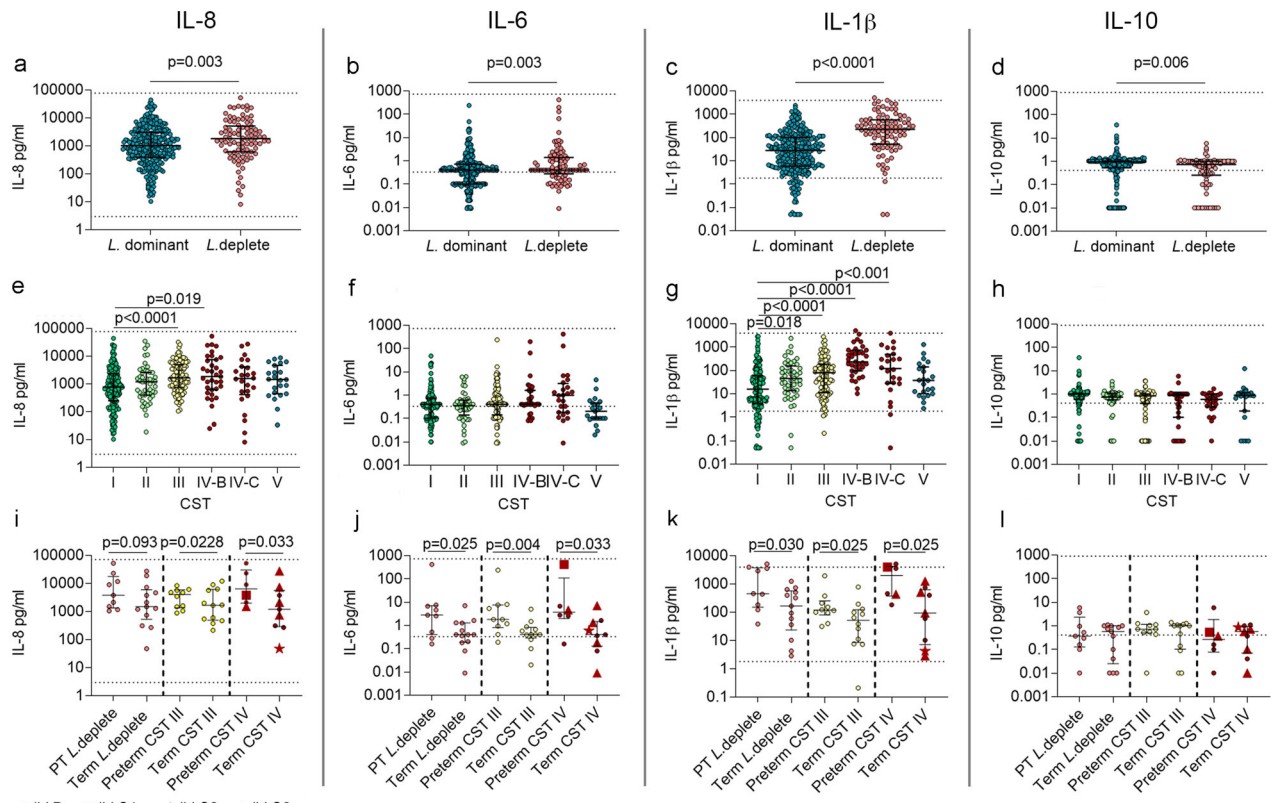

**Fig. 3 Cervicovaginal microbial composition modulates the local cytokines and leads to a pro-inflammatory response in women who deliver preterm.**
Cervicovaginal concentrations of IL-8 (**a**), IL-6 (**b**), IL-1β (**c**), and IL-10 (**d**) are shown from samples classed as *Lactobacillus* dominant or deplete,
$n = 385$ samples, $n = 133$ women. Statistical analysis was performed using a one-sided Mann–Whitney test. Cervicovaginal cytokine concentrations of IL-8
(**e**), IL-6 (**f**), IL-1β (**g**), and IL-10 (**h**), are shown relating to samples taken from a vaginal composition classed as CST I, II, III, IV-B, IV-C and V,
$n = 385$ samples, $n = 133$ women. Statistical analysis was performed using the Kruskal–Wallis and Dunn's multiple comparison's test. Cervicovaginal
concentrations of IL-8 (**i**), IL-6 (**j**), IL-1β (**k**), and IL-10 (**l**) were compared between women who delivered preterm or at term in those who were classed as
being *Lactobacillus* deplete ($n = 22$), or abundant in CST III (*L. iners*) ($n = 22$) or CST IV (diverse) ($n = 15$). CST IV was subdivided into CST IV-B ($n = 8$),
CSTIV-C1 ($n = 1$), CSTIV-C2 ($n = 1$) and CST IV-C3 ($n = 5$). Statistical analysis was performed using a one-sided Mann–Whitney test. Data are presented
as median values and interquartile ranges (25th and 75th percentiles). Source data are provided as a Source Data file.

lectin (MBL), IgG1 and IgG3 increased significantly between $12^{+0}$–$16^{+6}$ and $20^{+0}$–$24^{+6}$ weeks in women who delivered pre-term, whereas no statistically significant changes in any mediators of microbial recognition were seen between timepoints in women who delivered at term (Fig. 4a–f). A vaginal microbial composition that was *Lactobacillus* deplete was associated with higher concentrations of all mediators (Supplementary Fig. 3a–f). CST IV-B was associated with significantly higher concentrations of MBL, IgM and IgG1-4, whereas CST IV-C and CST III were only associated with statistically significant higher concentrations of IgM and IgG2-4. (Fig. 4g–l). By mid-pregnancy, in women who were *Lactobacillus* spp. deplete, preterm delivery was associated with significantly higher concentrations of MBL and IgM compared to women who delivered at term (Fig. 4m, n). Similarly, those who had CST IV and delivered preterm had higher concentrations of MBL and IgM (Fig. 4m, n), whereas CST III was associated with significantly higher concentrations of IgM (Fig. 4n) and IgG2-4 (Fig. 4p–r). Although CVF IgA and IgE were higher in association with *Lactobacillus* spp. depletion (Supplementary Fig. 4c, d), there were no significant differences between CST by delivery outcome (Supplementary Fig. 4g, h). Sample size limited the ability to compare outcome between sub-CST IV-B and IV-C, however, compositional data is presented in Supplementary Data 2. These data demonstrate a role for MBL, IgM, and IgG2-4 in determining delivery outcome in microbial-driven PTB.

**Complement activation, vaginal microbial composition, and preterm birth.** MBL activates the lectin pathway, while IgM/IgG immune complexes activate the classical pathway of the complement cascade. Given the increase observed in these mediators in women who delivered preterm with *Lactobacillus* depleted vaginal microbiota, or in association with CST III or CST IV, complement proteins C3b, C5 and C5a were measured in CVF. A significant increase in C3b and C5a was seen between $12^{+0}$–$16^{+6}$ and $20^{+0}$–$24^{+6}$ weeks in women who delivered preterm, whereas no significant changes were seen in women who delivered at term (Fig. 5a, c). C3b and C5 were increased in association with *Lactobacillus* depletion, regardless of outcome (Supplementary Fig. 4g–i). At species level, CST IV-B was associated with higher C3b and C5 concentrations (Fig. 5d, e) and CST IV-C was associated with higher concentrations of C3b (Fig. 5d). Concentrations of complement proteins were therefore compared between women who delivered preterm or at term in those who were *Lactobacillus* deplete, CST III or CST IV. Concentrations of C3b and C5 were significantly higher in women who were *Lactobacillus* deplete and delivered preterm, compared to women who delivered at term (Fig. 5g, h). In CST IV, C3b and C5 were significantly higher in women who delivered preterm compared to term, (Fig. 5g, h). Sample size limited the ability to compare outcome between sub-CST IV-B and IV-C, see Supplementary Data 2 for compositional data.

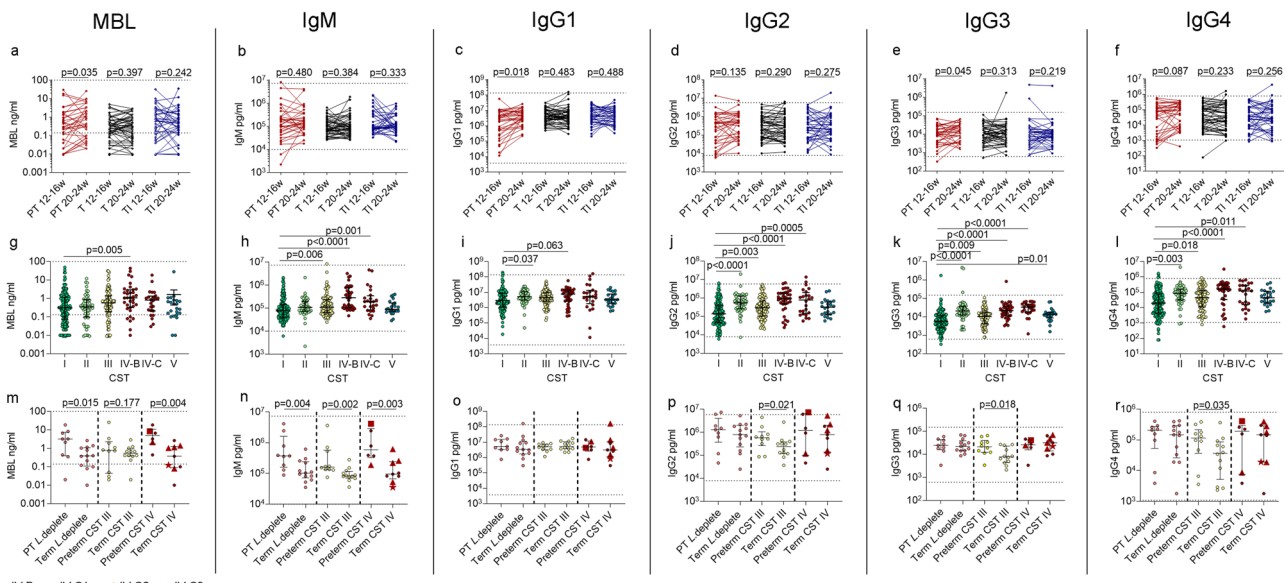

**Fig. 4 Cervicovaginal concentrations of mediators of microbial recognition differ depending on vaginal microbial composition and birth outcome.**
Cervicovaginal concentrations of MBL, IgM, and Ig1-IgG4 were measured and concentrations compared between 12–16 weeks and 20–24 weeks in women who delivered preterm, term without intervention, and at term following intervention **a–f**, $n = 122$. Statistical analysis was performed using a one-sided Wilcoxon matched pairs signed rank test. MBL, IgM, and IgG1-IgG4 concentrations were also compared between samples taken from women who were classed as CST I with CST II-V **g–l**, $n = 385$ samples from $n = 133$ women. The Kruskal–Wallis and Dunn's multiple comparison's test was used to determine statistical significance. Cervicovaginal concentrations of MBL, IgM, and Ig1-IgG4 were compared between women who delivered preterm or at term in those who were classed as being *Lactobacillus* deplete ($n = 22$), or abundant in either VMG 3 (*L. iners*) ($n = 22$) or VMG 4 (diverse) ($n = 15$). CST IV was subdivided into CST IV-B ($n = 8$), CSTIV-C1 ($n = 1$), CSTIV-C2 ($n = 1$) and CST IV-C3 ($n = 5$) (**m–r**). Statistical analysis was performed using a one-sided Mann–Whitney test. Data are presented as median values and interquartile ranges (25th and 75th percentiles). Source data are provided as a Source Data file.

A positive correlation was seen between concentrations of MBL (Fig. 6a–c) and IgM (Fig. 6d–f), and concentrations of CVF cytokines IL-8, IL-6 and IL-1β in women who were *Lactobacillus* deplete who delivered preterm. C3b correlated with concentrations of IL-8, IL-6 and IL-1β (Fig. 6g–i), and C5 correlated with IL-6 and IL-1β (Fig. 6j–l). This supports a mechanism whereby host–microbial-driven inflammation occurs via activation of the complement cascade in women who deliver preterm.

**Local immune milieu, vaginal microbial composition, and cervical shortening.** Comparison of the cervicovaginal immune milieu between women with ($n = 13$) and without ($n = 96$) cervical shortening between $12^{+0}–16^{+6}$ weeks of pregnancy showed increased levels IgM, C5, C5a and IL-6, and lower concentrations of IL-10 in those women with a short cervix (Fig. 7a–e). As microbial-driven inflammation is considered to be an aetiological factor for cervical shortening in a proportion of women, we next examined the interaction between microbial composition, immune mediators and cervical length. In women with a short cervix, there was an overrepresentation of CST III (*Lactobacillus iners*) (Fig. 7f) and increased relative mean abundance (Fig. 7g). This was associated with increased levels of IgM, IgG2, IgG4, complement proteins C3b, C5 and C5a, cytokines IL-6 and IL-1β, and lower concentrations of IL-10 (Fig. 7h–t) in women who developed cervical shortening during their pregnancy. While not all changes reached statistical significance (IgG2, IgG4, C5a and IL-1β, Fig. 7k, m, p, s), a trend in accordance with the hypothesis supports clinical significance. These results demonstrate that a short cervix is associated with local activation of the innate immune response, and that cervical shortening in the presence of *Lactobacillus iners* is also associated with activation of the adaptive immune response.

**The influence of progesterone and cervical cerclage on the local immune milieu.** The two most common interventions used in women at high risk of preterm birth who are found to have a short cervix in the second trimester of pregnancy are cervical cerclage and vaginal progesterone therapy. Forty women were sampled before and after the insertion of a cervical cerclage. Of these, 21 women received progesterone after insertion of the cervical cerclage and 19 women did not, according to patient or clinician preference. 14 women had braided suture material, and 25 women had monofilament, and in one case it was not possible to obtain information on suture material. Choice of cerclage material depended on either dual participation in the C-STICH trial (a multicentre randomised controlled trial to compare pregnancy outcomes between these two cerclage materials, https://www.isrctn.com/ISRCTN15373349), or was dependent on local standard practice. Nine of the women sampled pre- and post-cervical cerclage were C-STICH study participants, 5 were randomised to monofilament and 4 were randomised to braided suture material.

Local immune mediators were analysed pre- and post-cervical cerclage. There was no difference in cervicovaginal concentrations of mediators of microbial recognition, complement proteins, cytokines, or preterm birth rate between women who did or did not receive vaginal progesterone (Supplementary Fig. 5a–d, respectively). Cervical cerclage using braided material led to a significantly higher fold change in mediators of microbial recognition (Fig. 8a), complement proteins (Fig. 8b) and cytokines (Fig. 8c) following cerclage placement compared to monofilament. There was also a significantly higher preterm birth rate, 57% vs. 20%, in the braided group (Fig. 8d), compared to the monofilament group.

The influence of cerclage and cerclage material on immune mediators and vaginal microbial composition was examined

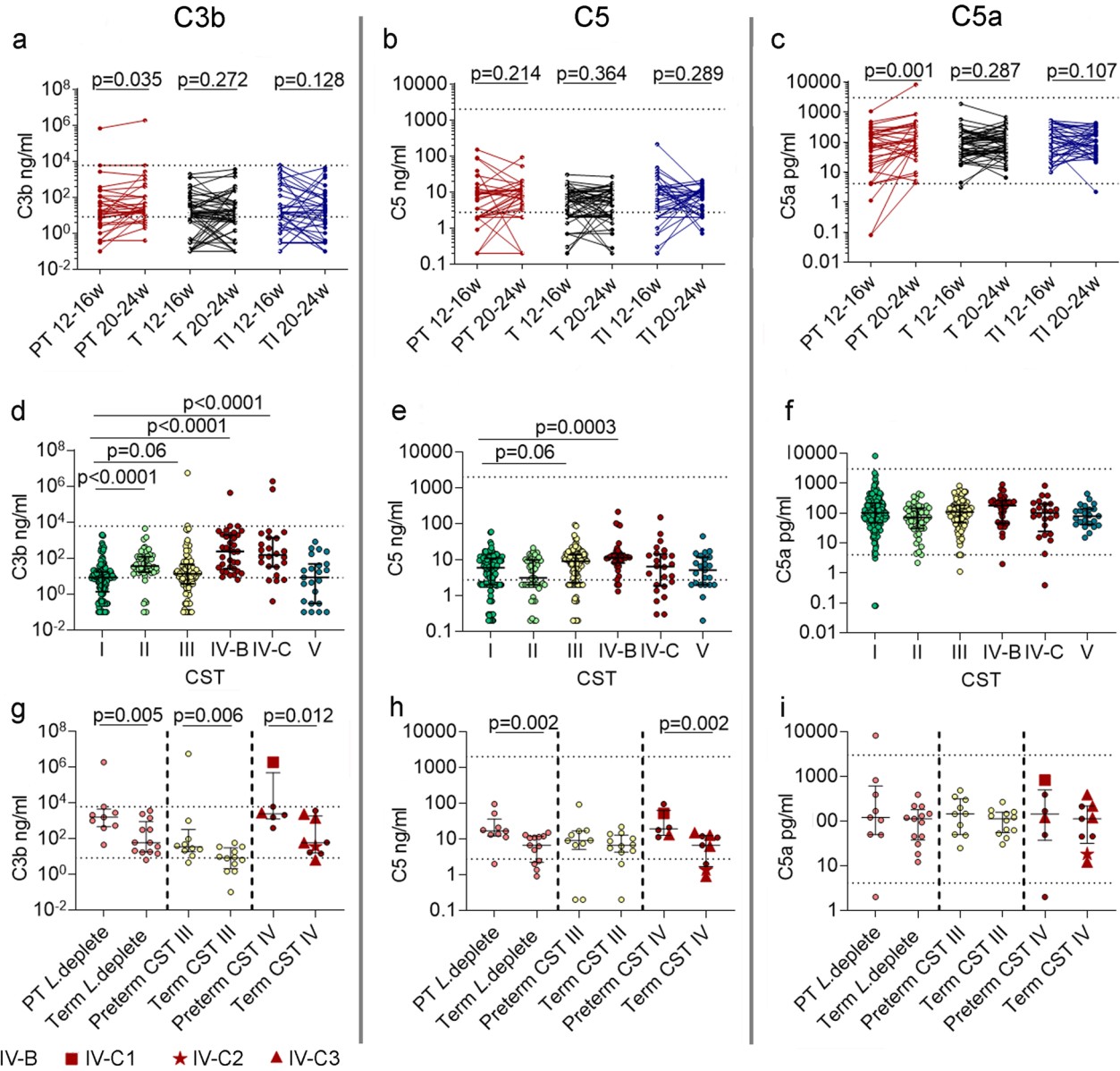

**Fig. 5 Vaginal microbiota activates complement in women who deliver preterm.** Cervicovaginal concentrations of C3b, C5, and C5a were measured and compared between 12–16 weeks and 20–24 weeks in women who delivered preterm, term without intervention, and at term following intervention (**a–c**), $n = 122$. Statistical analysis was performed using a one-sided Wilcoxon matched pairs signed rank test. C3b, C5, and C5a concentrations were also compared between samples taken from women who were classed as CST I with CST II–V (**d–f**) $n = 385$ samples from $n = 133$ women. The Kruskal–Wallis and Dunn's multiple comparison's test was used to determine statistical significance. Cervicovaginal concentrations of C3b, C5, and C5a were compared between women who delivered preterm or at term in those who were classed as being *Lactobacillus* deplete ($n = 22$), or abundant in either CST III (*L. iners*) ($n = 22$) or CST IV (diverse) ($n = 15$) (**g–i**). CST IV was subdivided into CST IV-B ($n = 8$), CSTIV-C1 ($n = 1$), CSTIV-C2 ($n = 1$) and CST IV-C3 ($n = 5$). Statistical analysis was performed using a one-sided Mann–Whitney test. Data are presented as median values and interquartile ranges (25th and 75th percentiles). Source data are provided as a Source Data file.

more closely in 33 women who delivered preterm who had longitudinal samples at timepoint A and B. Despite most women maintaining stability of microbial composition, there was a significant increase in local production of MBL, C3b, C5a, IL-8, IL-6, and IL-1β following a braided material cerclage (Fig. 9a–i). A degree of immune activation was seen in women who delivered preterm without intervention, with an increase in C3b, C5a and IL-6 (Fig. 9d, f, h, respectively). In contrast, monofilament insertion was not associated with immune activation (Fig. 9b–i). Of the 73 women who delivered at term with longitudinal samples between the A and B timepoint, five women had a cerclage using braided material, 15 had

monofilament and 54 women had no intervention (Supplementary Fig. 6a–i). The microbial composition was relatively stable in each group between timepoints (Supplementary Fig. 6). There was no immune activation seen in women who delivered at term without intervention, and monofilament also appeared to be immunologically inert (Supplementary Fig. 6b–i). An upward trajectory was seen between timepoint A and B in the concentrations of MBL, IgM, C5, C5a, IL-1β (Supplementary Fig. 6b, c, e, f, i) in most women who delivered at term with a braided cerclage material, but a statistically significant increase was seen only in C5a concentrations (Supplementary Fig. 6f).

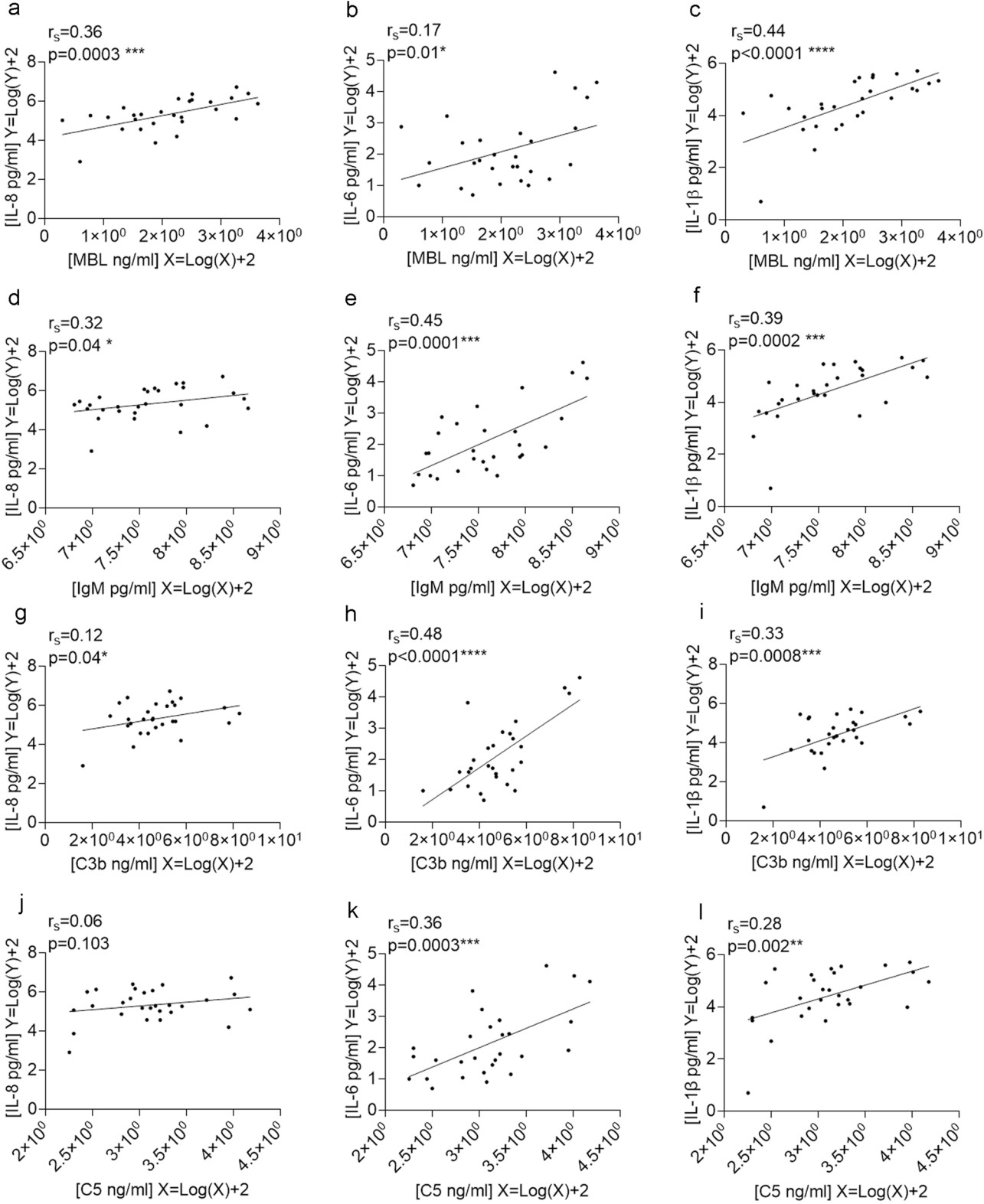

**Fig. 6 Cervicovaginal IgM and complement positively correlate with pro-inflammatory cytokines in women who are *lactobacillus* deplete and who deliver preterm.** In women who are *Lactobacillus* deplete and who deliver preterm a positive correlation was seen between cervicovaginal complement and cytokines. Concentrations were log transformed and a one-sided Spearman's correlation was performed between; MBL and IL-8 (**a**). IL-6 (**b**), IL-1β (**c**); IgM and IL-8 (**d**). IL-6 (**e**), IL-1β (**f**); C3b and IL-8 (**g**), IL-6 (**h**), IL-1β (**i**); C5 and IL-8 (**j**), IL-6 (**k**), IL-1β (**l**). n = 28 samples. Source data are provided as a Source Data file.

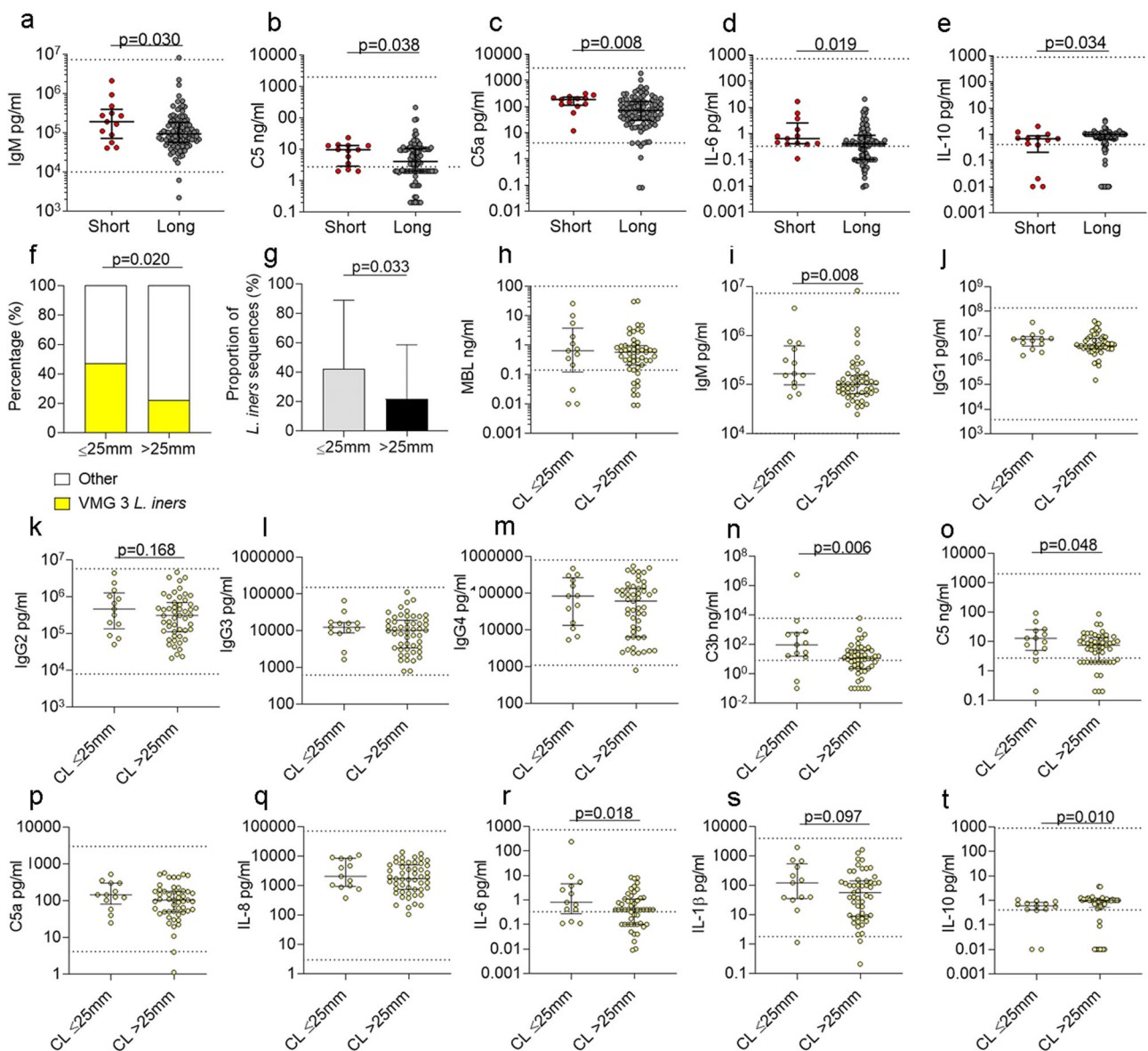

**Fig. 7 Cervical shortening is associated with activation of the adaptive and innate immune response, especially in the presence of *Lactobacillus iners*.**
Cervicovaginal immune mediators were analysed in women between 12–16 weeks gestation prior to intervention (**a**–**e**). Thirteen women had a cervical length (CL) ≤ 25 mm and 96 women had a cervical length >25 mm. Median concentrations of IgM (**a**), C5 (**b**), C5a (**c**), IL-6 (**d**), and IL-10 (**e**) are presented, $n = 109$ women. Statistical analysis was performed using a one-sided Mann–Whitney test. The percentage of women with a short cervix (≤25 mm) or long cervix (>25 mm) with *L. iners* at 12–16 weeks is presented in **f**, and the proportion of *L. iners* sequences is presented in **g**, $n = 111$ women. Immune mediators were analysed and compared between women who had L. iners (CST III) who developed cervical shortening (CL ≤ 25 mm) and women who maintained a CL > 25 mm. Mediators of microbial recognition included MBL (**h**), IgM (**i**), IgG1 (**j**), IgG2 (**k**), IgG3 (**l**), IgG4 (**m**), C3b (**n**), C5 (**o**), C5a (**p**), IL-8 (**q**), IL-6 (**r**), IL-1β (**s**), IL-10 (**t**), $n = 64$ samples. Statistical analyses were performed using a one-sided Chi square to compare proportions of VMG 3 between women with a CL ≤ 25 mm and >25 mm (**f**), and a one-sided Mann–Whitney to compare differences between sequence percentages and immune mediators (**g**-**t**), (**h**-**t**). Data are presented as median values and interquartile ranges (25th and 75th percentiles). Source data are provided as a Source Data file.

## Discussion

Our study identifies cross-talk between the host innate and adaptive immune response in microbial-driven cervical shortening and sPTB. To date, most studies exploring the local cervicovaginal immune milieu in the context of sPTB have focused on the innate immune response, and predominantly on the role of Toll-like receptors, cytokines, and chemokines. Elevated cervicovaginal concentrations of cytokines such as IL-8, IL-6 and IL-1β, TNF-α have been consistently reported in women who subsequently deliver preterm[20]. In our study of women at high risk of preterm birth, we demonstrate that this elevation occurs between

the early and late second trimester of pregnancy. Furthermore, we report on an increase in microbial mediators of recognition and complement proteins, demonstrating cross-talk between the innate and adaptive immune response in women who deliver spontaneously preterm.

Recent studies of the vaginal microbiome in pregnancy have linked increased microbial diversity and the presence of pathobionts with a higher risk of preterm birth[12,14,15,18,21–23], while the dominance of the vaginal niche by commensals such as *L. crispatus* reduces the risk[11,12,18,23,24]. In this study we used amplification of the V1/V2 hypervariable region for metataxonomic

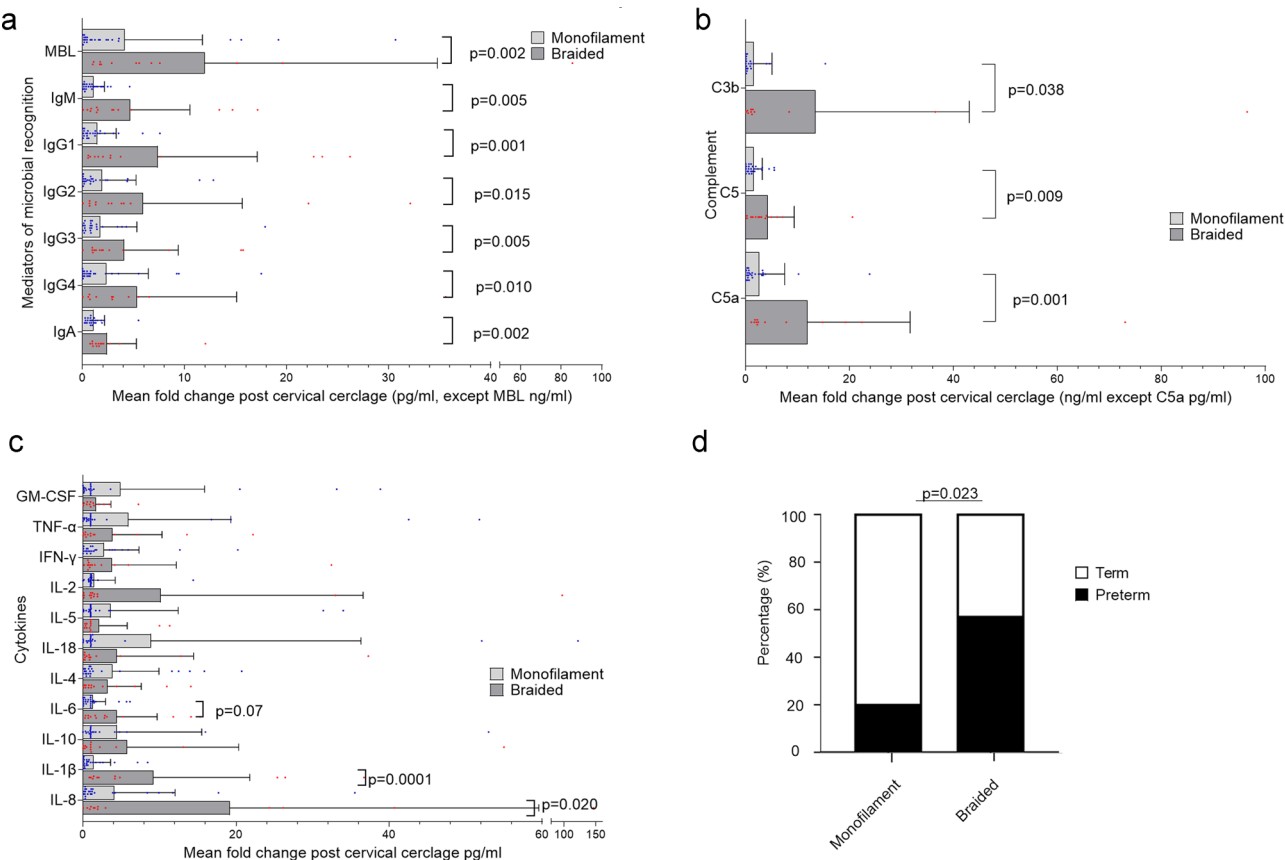

**Fig. 8 Braided cervical cerclage material is associated with immune activation and higher rates of preterm labour.** Cervicovaginal immune mediators were analysed from women pre- and post-cervical cerclage. The type of material was known for 39 women, 14 women received braided cerclage, and 25 women received monofilament. The fold change concentration of mediators of microbial recognition (**a**), complement proteins (**b**), and cytokine concentrations (**c**) between samples pre-cerclage and post-cerclage are compared between women who had braided cerclage ($n = 14$) and women who had monofilament cerclage ($n = 25$). The percentage of women who delivered preterm with a braided cerclage was compared with the percentage who delivered preterm following a monofilament cerclage (**d**). Statistical analysis was performed using a one-sided Mann–Whitney test and one-sided Fisher's exact test. Data are presented as mean and standard deviation. Source data are provided as a Source Data file.

analysis and clustered the resulting sequence data using the VALENCIA classifier. Amplification of the V1/V2 regions has been widely used in the study of the vaginal microbiome, including the original Ravel community state type classification study[25]. This approach has the advantage over application of other regions of improved discrimination between species of *Lactobacilli*, which are the most prevalent genus in the vaginal microbiota. It has the disadvantage of that if a universal forward primer is used, mismatches can lead to under representation of important vaginal bacterial genera including *Gardnerella*. However, as we have done in this study, this problem can be overcome through the use of a mixed formulation of the 27F forward primer, which has been shown to maintain the rRNA gene ratio of *Lactobacillus* spp. to *Gardnerella* spp[26].

Taxonomic profiles of vaginal microbiota communities are commonly sorted into discrete categories termed community state types or vaginal microbiome groups (VMGs) based on the results of hierarchical clustering of the pairwise comparisons. As the results are dependent on the particular population that was analysed, this approach makes cross study comparisons challenging. The VALENCIA algorithm is a nearest centroid-based tool that works by classifying samples based on their similarity to a set of reference centroids defined against a set of 13,160 taxonomic profiles from 1975 women of reproductive age. This approach has been validated and tested in multiple different ethnic populations and is considered a robust method for unbiased, reproducible,

and standardised reporting of vaginal community state types[27]. The VALENCIA pipeline characterises individual subject vaginal bacterial communities into seven CSTs of which three represent *Lactobacillus* spp. deplete communities (CST IV-A, IV-B and IV-C), and thirteen sub-CSTs. Most of our *Lactobacillus* spp. deplete samples were of IV-B or IV-C3 subtype, with minimal or no samples classed as IV-A, IV-C0-2, and IV-C4. We therefore analysed our data using the seven community state type classification.

Moving beyond associative studies of the vaginal microbiota and preterm birth to gain mechanistic insight has been hampered by the lack of robust in vitro and in vivo model systems[28]. Our data, using a carefully designed human study, indicate that the combination of a high-risk vaginal microbial composition and specific maternal immune response modulates the risk of sPTB. We have previously shown that a dysbiotic vaginal microbial composition is associated with higher concentrations of pro-inflammatory cytokines during pregnancy[13], similar to the pro-inflammatory milieu that occurs in non-pregnant women in the presence of dysbiosis[29]. Fettweis et al.[12], demonstrated a positive correlation between cervicovaginal pro-inflammatory cytokines and dysbiotic taxa in women who delivered preterm whereas a negative correlation was seen between *L. crispatus* and cytokine concentrations. In our study, we found a significant increase in cervicovaginal concentrations of IL-8, IL-6 and IL-1β in women who were deplete of *Lactobacillus* spp., or in association with a

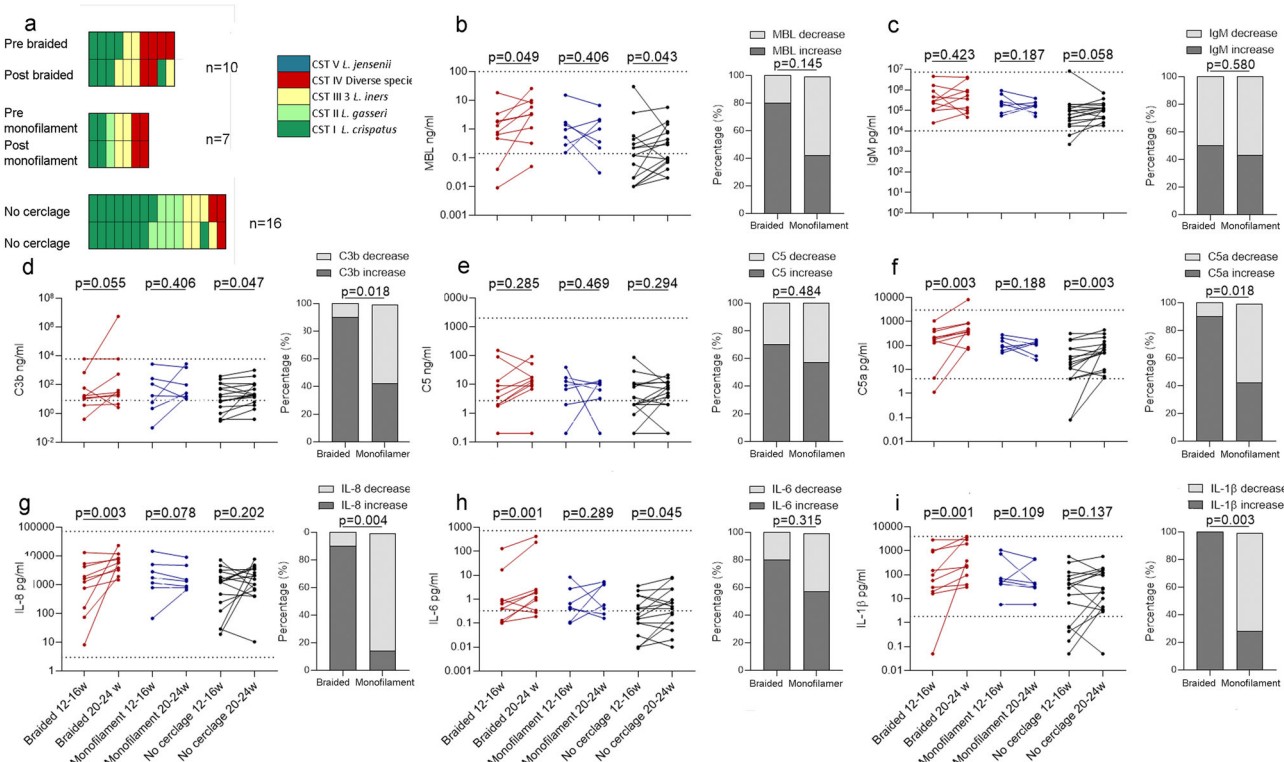

**Fig. 9 Cerclage using braided material is associated with activation of complement and cytokine production in women who deliver preterm.** Vaginal microbiota composition (**a**) and cervicovaginal immune mediators (**b**–**i**) were analysed in 34 women who delivered preterm between 12–16 weeks and again between 20–24 weeks. Ten women had a cerclage with braided material and 7 women had a cerclage with monofilament material between timepoints, whereas 16 women had no intervention (**a**). The concentration of MBL (**b**), IgM (**c**), C3b (**d**), C5 (**e**), C5a (**f**), IL-8 (**g**), IL-6 (**h**), and IL-1β (**i**) were compared between 12–16 and 20–24 weeks in women who had a braided cerclage, monofilament cerclage and no cerclage. The proportion of women who had an increase in immune mediators were also compared between suture material. Statistical analysis was performed using a one-sided Wilcoxon matched pairs signed rank test and a one-sided Fisher's exact test, N = 33 women. Source data are provided as a Source Data file.

diverse vaginal microbial composition. We also found that *L. iners* was associated with higher concentrations of IL-8 and IL-1β. However, the highest cytokine concentrations were seen in those women with these microbial signatures who subsequently delivered preterm, suggesting excessive innate immune response to specific vaginal microbiota as a potential mechanism leading to untimely activation of parturition in these women. For the primary analysis of this study, *Lactobacillus* spp. deplete community states were combined to IV-A, B or C, (with CST IV-C0-4 combined within CST IV-C). Separate analyses were not feasible between CST-IV subtypes due to the limitations in sample number. While our study clearly supports the association between *Lactobacillus* spp. depletion and inflammation in women delivering preterm, further larger studies will be needed to explore the effects of each of the *Lactobacillus* spp. deplete subtypes.

The mechanism by which vaginal microbiota induce inflammation and trigger parturition is poorly understood. Variation in the maternal host immune recognition and response to vaginal microbiota would explain why only a proportion of women with a high-risk vaginal microbial profile deliver preterm. MBL binds mannose, *N*-acetylglycosamine and fucose carbohydrate moieties on microbial surfaces, and facilitates microbial elimination via opsonisation, phagocytosis, and inhibition of bacterial propagation[30]. In our study we saw an increase in CVF MBL concentration in women who delivered preterm, with the increase occurring between early and late second trimester. MBL concentrations were elevated in those who were *Lactobacillus* deplete and with a diverse microbial composition. The most likely source of CVF MBL is vaginal epithelial cells[31], and its production may

be differentially regulated at the cervicovaginal interface in comparison to the systemic release by the liver. Although studies of cervicovaginal MBL in pregnancy are lacking, CVF MBL is increased with vulvovaginal candidiasis[32], and binding has been shown with *G. vaginalis* in non-pregnant women[33], supporting its role in the recognition of pathogenic organisms.

Immunoglobulins are the humoral component of the adaptive immune response, which serve to neutralise microbes and facilitate opsonisation for targeted cell death. In non-pregnant women, B cells account for a small proportion of the immune cells at the cervicovaginal interface but are most abundant in the ectocervix compared to the endocervix, a site of greater bacterial load[34,35]. B cell and immunoglobulin concentrations are hormonally regulated with menstrual cycle fluctuations and changes occurring following the menopause[36]. Our study revealed that the most abundant immunoglobulin in CVF is IgG followed by approximately equal concentrations of IgA and IgM in pregnancies that went to term. This is consistent with data from studies of cervical mucous immunoglobulins in healthy pregnancies[37,38], although only one study has reported on IgM concentrations and found lower concentrations than IgA at term[38].

We found no changes in IgA across gestational age, or by CST in relation to pregnancy outcome, implying that IgA has no functional role in microbial-driven preterm birth. In contrast, concentrations of IgM were higher in women who delivered preterm compared to at term and was further increased in women who delivered preterm with a vaginal microbial composition dominated by *L. iners* or with *Lactobacillus* depletion. IgM has a

pentameric structure, which allows for higher valency compared to other immunoglobulins, thus rendering it 100–10,000 more effective at agglutination than IgG. IgM is in both natural and immune (secreted in response to antigen exposure) form, with the former containing more flexible intrinsic antigen binding sites to facilitate broader interactions with a variety of antigens[39].

Although IgG1 increased during pregnancy in women who delivered preterm, there was no significant difference in concentrations in relation to vaginal microbial composition. In contrast, IgG2 and IgG3 increased in association with *Lactobacillus* depletion. In women who were abundant in *L. iners*, those who delivered preterm had significantly higher concentrations of IgG2, 3 and 4 compared to those who delivered at term. IgG1 responses are primarily to protein antigens, whereas the response to bacterial capsular polysaccharide antigens are mostly IgG2 mediated[40]. IgG3 has the highest efficacy for complement activation and has significant pro-inflammatory effector functions and is typically increased in response to viral infection. In contrast, IgG4 is the only subclass that cannot bind C1q to activate complement. It is unsurprising therefore to see striking differences in cervicovaginal IgG2 and 3, particularly given that our data suggest microbial-driven complement activation, and the emerging evidence for the role of viruses in modulating microbial host responses and preterm labour[41,42]. Future work should focus on determining the degree of reactivity of the Ig classes and subclasses to specific vaginal microbiota to establish if this influences the local immune milieu and ultimately the risk of preterm birth.

The complement system is an integral component of the immune response and bridges the adaptive and innate response to pathogens. MBL activates the lectin pathway, whereas IgM and IgG1-3 immune complexes activate the classical pathway (Fig. 10a). Both pathways converge on the central component C3, which is activated via C3 convertase. Upon activation C3 is cleaved to generate C3b and C3a. C3b binds to C3 convertase to form C5 convertase, which leads to release of C5b and C5a.

C3a and the more potent C5a lead to increased vascular permeability and chemoattract phagocytes, and C3b opsonises pathogens and aids phagocytosis via C3b receptors. The alternative pathway also activates a positive feed forward amplification loop via C3b deposition on the surface of pathogens. Dysregulation of the complement system has been implicated in preterm birth in humans, yet analysis of complement proteins has been limited to plasma and amniotic fluid in poorly phenotyped cohorts with likely mixed aetiology[43–45]. In our study, we show that CVF C3b and C5a increased during pregnancy in women who delivered preterm. C3b and C5 concentrations were increased when the vaginal microbial composition was deplete of *Lactobacillus* or of high diversity, with highest concentrations seen in women who went on to deliver preterm. Moreover, MBL, C3b, C5 and C5a positively correlated with the pro-inflammatory cytokines IL-8, IL-6 and IL-1β, demonstrating a pathogenic role for the complement pathways at the cervicovaginal interface in microbial-driven inflammation and preterm labour.

Cervical shortening is a risk factor for preterm birth. We demonstrate that cervical shortening as early as 12–16 weeks and prior to any intervention, was associated with higher concentrations of IgM, C5, C5a, IL-6, and lower concentrations of IL-10. Consistent with our previous study[11], we showed that the most prevalent microbial signature in women who developed a short cervix was a predominance of *L. iners*. Furthermore, those with *L. iners* who developed cervical shortening had significantly higher concentrations of IgM, C3b, C5, and IL-6. Murine studies using intravaginal LPS also demonstrate a role for C3 and C5a in cervical ripening and preterm birth[46,47]. It is plausible that defragmentation of C3 and C5 leads to C3a and C5a-driven chemoattraction of neutrophils to aid phagocytosis with the assistance of C3b (Fig. 10a). Neutrophils are a major source of prostaglandins, matrix metalloproteinases, and cytokines, which then mediate cervical remodelling and dilation[48,49].

Finally, we examined the potential influence of the two most commonly used therapeutic interventions on the cervicovaginal

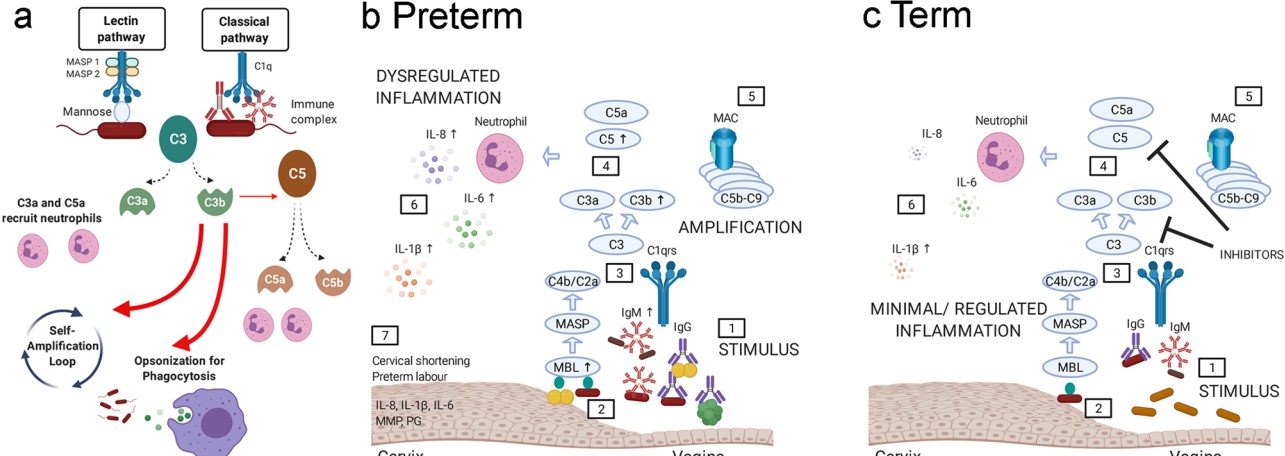

**Fig. 10 Proposed mechanism for microbial-driven preterm birth.** The complement system is an integral component of the immune response and bridges the adaptive and innate response to pathogens. MBL activates the lectin pathway, whereas IgM and IgG1-3 complexes activate the classical pathway (**a**). Both pathways converge on the central component C3, which is activated via C3 convertase. Upon activation C3 is cleaved to generate C3b and C3a. C3b binds to C3 convertase to form C5 convertase, which leads to release of C5b and C5a. C3a and the more potent C5a lead to increased vascular permeability and chemoattract phagocytes, and C3b opsonises pathogens and aids phagocytosis via C3b receptors. The alternative pathway also activates a positive feed forward amplification loop via C3b deposition on the surface of pathogens. We propose that in women who deliver preterm, CST III (*L. iners*) and CST IV (diverse species) activate the complement cascade, which leads to a local pro-inflammatory immune milieu. It is likely that the presence of activated neutrophils leads to an increase in the local concentration of IL-8, IL-6, and IL-1β, prostaglandins (PGs) and matrix metalloproteinases (MMPs). If a dysregulated response occurs, this may lead to cervical shortening and preterm labour (**b**). In women who deliver at term (**c**), there is a predominance of CST I (*L. crispatus*). However, if CST III (*L. iners*) or CST IV (diverse species) is seen, we propose that a more regulated immune response prevents cervical shortening and the triggering of early parturition. Figure created with BioRender.com.

immune milieu. Progesterone has been shown to have anti-inflammatory properties[50], however, we did not see evidence of immunomodulation, with no differences in the concentrations of immunoglobulin, complement or cytokine production in women who received progesterone. This is consistent with previous data that showed no anti-inflammatory effect of progesterone in women who had cervical shortening[51], and with progesterone having no effect on the vaginal microbiota[11]. In contrast, the mean fold change in immunoglobulin, complement and pro-inflammatory cytokines following cervical cerclage with the braided suture material was significantly higher than following cerclage with monofilament. Furthermore, the sPTB rate was significantly higher in women who received a cervical cerclage with braided suture material. This is consistent with our previous multicentre case–control analysis, which showed higher rates of PTB, and in a smaller randomised group, a shift towards dysbiosis at 4 weeks and increases in pro-inflammatory cytokines following cerclage with braided suture material compared to monofilament[13]. In the present study we did not see a significant change in vaginal microbial composition following insertion of either material, however, our sampling following cerclage was later, and we may have missed any transient change. Nevertheless, in women who delivered preterm despite having a cervical cerclage, there was a significant increase in MBL, C3b, C5a, IL-8, IL-6, and IL-1β following braided cerclage insertion, but no immunomodulatory effect was seen in those who received monofilament. In contrast, in those women who delivered at term, only C5a was significantly increased in women who received a cerclage with braided suture material. This suggests that in women who deliver preterm following braided cervical cerclage, other local factors have additive or synergistic effects that lead to a more substantial dysregulation of the environment. In our study, we show that monofilament suture is immunologically inert, has better outcomes, and therefore is recommended as the material of choice for cervical cerclage.

In summary, we propose a mechanism whereby the maternal host immune response to vaginal microbiota drives inflammation, cervical shortening, and preterm labour (Fig. 10b). Although local inflammation and/or a high-risk vaginal microbial composition does not always lead to preterm birth (Fig. 10c), we propose that a dysregulated innate and adaptive immune response, bridged by the complement cascade, is what triggers microbial-driven preterm parturition. We have demonstrated that the commensal *L. crispatus* does not activate inflammation, and may be protective against inflammation, which provides a rationale for its use as a vaginal live biotherapeutic product to reduce preterm birth rates. We have identified the complement system as a potential key player in driving the adverse host–microbial interactions, which provides the rationale for the potential application of complement therapeutics[52], especially if delivered locally. In conclusion, we have proposed a potential mechanism involving cross-talk between the innate and adaptive immune response in microbial-driven preterm birth, which could lead to the development of novel therapies for the prevention of preterm birth.

## Methods

**Study design.** Study participants were prospectively recruited from preterm birth prevention clinics from five UK hospitals (Chelsea Westminster Hospital, Edinburgh Royal Infirmary, St Marys Hospital London, Queen Charlottes Hospital, and University College London Hospital) between February 2016 and June 2018. The study was approved by the National Health Service, National Research Ethics Committee in Stanmore, London (REC 14/LO/0328). Participants provided written informed consent. Inclusion criteria were history of previous sPTB, previous mid-trimester loss, recurrent miscarriage, incidental finding of cervical shortening and/or cervical excisional treatment. Exclusion criteria were women under the age of 18, HIV or hepatitis C positive status, and vaginal intercourse or bleeding within 72 h of sample collection. All pregnant women in the UK are screened for syphilis. Otherwise, routine screening for sexually transmitted diseases was not included in the study protocol but would be undertaken in any woman with signs or symptoms. Women were advised to avoid vaginal douching at their first appointment. Reporting of clinical data complies to the STROBE guidelines.

Women were recruited at timepoint A ($12^{+0}$–$16^{+6}$ weeks) and seen at two further study timepoints B ($20^{+0}$–$24^{+6}$ weeks) and C ($30^{+0}$–$34^{+6}$ weeks) unless delivery occurred before. At each timepoint, CVF was sampled using a BBL™ Culture Swab™ MaxV Liquid Amies swab (Becton, Dickinson and Company, Oxford, UK). Swabs were placed on ice immediately and stored at –80 °C until use. Use of all human material was approved by the National Health Service, National Research Ethics Committee in Stanmore, London (REC 14/LO/0328). At each visit cervical length was measured by transvaginal ultrasound in women with an empty bladder. Metadata was collected at each visit and stored on a secure database. This included maternal age, BMI, ethnicity, past obstetric history, medical history, any interventions for preterm birth including progesterone and cervical cerclage. Outcome data was collected following delivery. Women were grouped by delivery outcome, spontaneous preterm birth with or without PPROM (iatrogenic preterm births were not included in the study), term uncomplicated (without intervention), and term intervention (women who had either cerclage and /or progesterone).

**Intervention.** Study participants may have required a clinical intervention such as progesterone and/or cervical cerclage to prevent preterm birth. This included women with cervical shortening on transvaginal ultrasound, defined as a cervical length ≤ 25 mm, and women with a history of cervical insufficiency and/or previous cervical cerclage. Where possible, cervical length was measured prior to and after the clinical interventions. Cerclage material was either monofilament or braided. The choice of cerclage material was based on either randomisation to C-STICH (REC: Cambridgeshire and Hertfordshire, ISRCTN 15373349) or according to local practice. 10 of the 59 (17%) women who received a cervical cerclage were dual participants of the C-STICH study.

**Luminex immunoassays.** The BBL™ CultureSwab™ was thawed on ice. Supernatant was extracted from the sponge of the swab using a sterile syringe and pressure to release a volume of ~350 μl. This was centrifuged at $3000 \times g$ for 10 min. Protease inhibitor (5 μl/ml; Sigma-Aldrich), was added to the supernatant. This was used for Luminex® immunoassays to quantify the chosen cytokines, complement and immunoglobulin analytes (see Supplementary Table 4 for Immunoassay kit catalogue numbers and source). Results were expressed in picograms or nanograms per ml of cervicovaginal fluid volume. IL-8 was measured on a single plex Human Premixed Analyte Kit (R&D Systems/Bio-Techne), following a tenfold dilution using Calibrator Diluent RD6-52. The remaining cytokines, IL-1β, IL-2, IL-4, IL-5, IL-6, IL-10, IL-18, IFN-γ, GM-CSF and TNF-α, were measured on a multiplex plate, Human Premixed Multi-Analyte Kit (R&D Systems/Bio-Techne) and no dilution was required. Complement analytes, C5, C5a and MBL were detected using the Human Complement Magnetic Bead panel 1 (Milliplex® Merck/Millipore), and Human Complement Magnetic Bead panel 2 (Milliplex® Merck/Millipore) was used to detect C3b/iC3b. Dilution of the CVF supernatant was not required for the complement assays. The ProcartaPlex Human Antibody Isotyping panel 7-plex (Thermofisher) was used to detect IgM, IgG1, IgG2, IgG3, IgG4, IgA and IgE. IgG1 required a 50-fold dilution with Universal Assay Buffer included in the assay kit. Immunoassays were run together with the Bio-Plex 200 system (Bio-Rad Laboratories Ltd). All samples were analysed on 96-well plates in duplicates. The coefficient of variation was calculated across all plates for standards and a common pooled sample analysed on all plates to assess for plate-to-plate variation.

**DNA extraction and 16S rRNA sequencing.** DNA extraction from the BBL™ CultureSwab™ was performed as previously described[53]. The V1-V2 hypervariable regions of 16 S rRNA genes were amplified using the following primers. The forward primer set (28f-/YM) consisted of a mixture of the following primers mixed to a 4:1:1:1 ratio; 28F-Borrellia GAGTTTGATCCTGGCTTAG; 28F- Chlorlex GAATTTGATCTTGGTTCAG; 28F- Bifido GGGGTTCGATTCTGGCTCAG; 28F-YM GAGTTTGATCNTGGCTCAG. This mixed formulation of the 27 F forward primer (27F-YM) has been shown to maintain the rRNA gene ratio of *Lactobacillus* spp. to *Gardnerella*[26]. The reverse primer consisted of was 388 R GCTGCCTCCCGTAGGAGT. Sequencing was performed on an Illumina MiSeq platform (Illumina Inc.) at Research and Testing Laboratory, (RTL Genomics) in Lubbock, Texas, USA.

To calculate microbial abundance the 16S amplification primers were trimmed using Cutadapt[54] followed by quality control using FastQC[55]. Amplicon sequence variant (ASV) counts per sample were calculated using the Qiime2 pipeline[56], where we used DADA2[57] for denoising. ASVs were taxonomically classified to species level using the STIRRUPS reference database[58]. Counts were normalised with relative log-expression using DESeq2[59]. Measures of microbial diversity: inverse Simpson and Shannon indexes and richness (species observed) were calculated using the Phyloseq package in R[60]. For analyses of the vaginal microbial composition, at genus level, samples were grouped into those that were *Lactobacillus* dominant (≥75% *Lactobacillus* species), or *Lactobacillus* deplete, (<75% *Lactobacillus* species). At species level, samples were grouped using the VALENCIA centroid classification programme[27] into seven community state types, four of which were dominated by a single species of *Lactobacillus*:CST I (*L. crispatus*), CST II (*L. gasseri*), CST III (*L. iners*), and CST V (*L. jensenii*), and three

which were not; CST IV-A, CST IV-B and CST IV-C. CST I was further divided to I-A and I-B sub-CST, CST III was further divided to III-A and III-B sub-CST, and CST IV-C was further divided to IV-C0, IV-C1, IV-C2, IV-C3, and IV-C4 sub-CST. Heatmaps were used to visualise the abundance data - hierarchical clustering with Ward linkage of the top 20 most abundant species.

**Statistical analysis.** Statistical analyses were performed using Graphpad Prism 9.0.0. Gestational age changes in cervicovaginal cytokines, complement and immunoglobulins were compared across the three sampling timepoints using the Kruskal–Wallis test and Dunn's post hoc multiple comparisons test. Gestational age changes were compared across matched samples between two sampling timepoints using Wilcoxon matched pairs signed rank test. Comparisons of the inverse Simpson index and microbial richness were made using the Mann–Whitney $U$-test. Differences in the immune profiles in *Lactobacillus* dominant and deplete samples, and between vaginal microbial groups were analysed using the Mann–Whitney $U$-test or the Kruskal–Wallis test and Dunn's post hoc multiple comparisons test. Spearman's correlation was used to compared immune analytes in women with preterm deliveries with *Lactobacillus* depletion. Immune mediators in the presence of normal cervical length and cervical shortening, and pre- and post-cervical cerclage were analysed using the Mann–Whitney $U$-test. The Fisher's exact test was used to compare the change in concentration of immune mediators following monofilament and braided cervical cerclage. For all tests, the level of statistical significance was taken as a $p$-value $\leq 0.05$.

**Reporting summary.** Further information on research design is available in the Nature Research Reporting Summary linked to this article.

## Data availability
Source data are provided with this paper. Source Data File includes the raw data used to create each figure. Supplementary Data 1 includes metadata including ENA accession numbers, All sequence data is available on the ENA database browser https://www.ebi.ac.uk/ena. Source data are provided with this paper.

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

## Acknowledgements

We would like to thank all women who have participated in this study and members of the Women's Health Research Centre who facilitated and coordinated study recruitment and sample collection. We would also like to acknowledge our Parents as Partners public involvement group for contributing to study design and interpretating the results from the perspective of parents who have experienced preterm birth. This work was funded by the March of Dimes European Preterm Birth Research Centre at Imperial College London and supported by the National Institute of Health Research (NIHR) Imperial Biomedical Research Centre (BRC), NIHR Clinical Lectureship Scheme, and the Genesis Research Trust. A.L.D. is supported by the UCLH NIHR Biomedical Research Centre. S.J.S. is funded by a Wellcome Trust Career Development Fellowship (209560/Z/17/Z).

## Author contributions

L.S., D.C., P.R.B., and D.A.M. conceptualised the study and developed the experimental design. Clinical sampling and coordination of metadata was performed by D.C., M.A., R.B., A.D., H.V.L., J.N., S.S., L.S., V.T., T.G.T., and P.R.B. Experiments were performed by D.C., S.A., B.M., and Y.S.L. Data processing, analysis, and interpretation was performed by D.C., D.A.M., L.S., S.K., P.R.B., P.K., and M.B. D.C. and L.S. prepared all figures and tables. D.C. and L.S. wrote the first draft of the manuscript. All authors critically reviewed, read and approved the final manuscript.

## Competing interests

The authors declare no competing interests.
