## [Peer Review File · Nature Communications]

Microbial-driven preterm labour involves crosstalk between the innate and adaptive immune responseREVIEWER COMMENTS

Reviewer #1 (Remarks to the Author):

Here the authors assessed the impact of the vaginal microbiome, inflammation, and complement in women at high risk for spontaneous preterm birth (sPTB). They recruited 132 women by parameters associated with sPTB from 5 UK hospitals and collected swabs (for microbiome) and CVL (for inflammation). Of high importance, they found very compelling associations with increased inflammation (IL-6, IL-1b and IL-8) that was associated with preterm birth, as well as increased Igs, complement pathway proteins. and mannose binding lectins. Importantly this was also associated with a non-lactobacillus vaginal microbiota (aka bacterial vaginitis). They also assessed the impact of 2 major treatments, either progesterone or cervical cerclage, but didn't find major changes to immunity or complement. This is an extremely important study and may lead to potential targets for decreasing sPTB.

Comments:

1. The authors find changes in Ig subclasses relative to sPTB however it is unclear what these Abs are to. It would greatly benefit the paper to have insights into what these Abs are specific to.
2. The clustering they used for vaginal communities is not standard, and really not what one would expect in the vaginal microbiome. This is likely because they used V1/V2 instead of V3/V4, which gives more Lactobacillus species identification but less overall genera, and specifically under representation of extremely important and common bacteria such as Gardnerella. If possible, redoing all 16S with V3/V4 is ideal. If this isn't possible this needs to be extremely carefully addressed in the manuscript and more standard vaginal microbiome community groups relative to inflammatory factors and SPTB needs to be re-analyzed.
3. Figures 2, 4 and 5 a-f are difficult to interpret. Use of better labeling would be helpful
4. Some of the figure colors are very difficult to see between, and if printed in black and white would be near impossible to see differences, such as green colors in figure 7. But through the authors should address this.

Reviewer #2 (Remarks to the Author):

The manuscript contains a lot of data about infection and the immune response in some preterm labor. The assumption that preterm labor is related to infection has been around a while and is generally accepted. That the infection is bacterial in origin has not been shown and the fact that this manuscript does not mention other possible infectious agents makes the data interesting but less important. The figures in general show that almost every cytokine, immunoglobulin or immune component has a statistical link suggests equally that none of them do or that the statistics used in comparison are not applicable in this instance. The data suggests that the Wilcoxon test may not be appropriate for this data set. The size of the figures and the fact no number are mentioned in the text are off putting.

While the authors suggest that a cross talk is apparent in the study no mechanisms or hypothesis is suggested. The grouping of bacterial agents is compelling but the lack of focus and the number of data points make it difficult to read. A more focused concise report is warranted and an hypothesis driven report warranted

Reviewer #3 (Remarks to the Author):

Summary: This manuscript addresses an important topic of high public health relevance. Leveraging a cohort of high-risk pregnant women for spontaneous pre-term birth (sPTB) they showed that maternal host immune response to vaginal microbiota drives inflammation, complement activation, cervical shortening and preterm labor. Specifically, 133 pregnant women with previous sPTB, previous PPRM, previous mid trimester pregnancy loss, previous large loop excision of the transformation zone of the cervix, or a combination of these were prospectively evaluated from early pregnancy (week 16) till delivery. Demographic and clinical profiling including timing of delivery and cervical length by transvaginal

ultrasound were performed. Vaginal swabs were obtained for microbiota profile using 16S rRNA sequencing, and for assays of inflammatory cytokines, complement and immunoglobulin analytes. Overall, the study was well designed, and pertinent predictor and outcome variables were captured to address the key research questions. Addressing the following critiques will significantly enhance the manuscript.

1. The control group include participants with risk factors for sPTB who did not experience pre-term delivery – it could be argued that a better control group should be pregnant women with no sPTB risk factor. This control used in the current study could have biased the outcomes and impacted the effect size as they were selected from a population of at-risk individuals for the main outcome of the study.
2. The study excluded participants with HIV and viral hepatitis but did not screen for nor exclude those with sexually transmitted infections including GC, CT, TV, etc. It is clearly established that STIs could alter the vaginal microbiota and also independently induce inflammatory changes and complement activation in manners that may alter the outcomes measured in the study.
3. The target population – women of reproductive age group, though pregnant are likely to have diverse vaginal hygiene practices that could affect the composition of the vaginal microbiome – were history of vaginal hygiene captured?
4. While the observed differences for some of the outcomes (VMG, cytokines, complements, immunoglobulins) were impressive, the absences of differences in others raised the question of power. Was the study powered a priori for these outcomes? If so, could sample size justification be provided in the statistical section?
5. A major concern with vaginal cytokine assays using cervicovaginal secretions is the accuracy of the concentration measures. The volume of secretions absorbed by each swab is very likely to be different and may thus influence the estimation of the amount of analytes present – the manuscript in its current form did not address this issue in the methods section.

15th of June 2021

Dear Reviewers,

We would like to resubmit the following manuscript to the Nature Communications.

Microbial driven preterm labour involves crosstalk between the innate and adaptive immune response

Ref: Nature Communications manuscript NCOMMS-21-12329

Thank you for taking the time to review this revised manuscript for publication in Nature Communications. We have addressed each of the comments point by point below, and have made amendments to the manuscript text, figures, and supplementary material. We feel these amendments have improved the quality of the manuscript as a result.

Best wishes,

Dr Lynne Sykes

Reviewer #1 (Remarks to the Author):

Here the authors assessed the impact of the vaginal microbiome, inflammation, and complement in women at high risk for spontaneous preterm birth (sPTB). They recruited 132 women by parameters associated with sPTB from 5 UK hospitals and collected swabs (for microbiome) and CVL (for inflammation). Of high importance, they found very compelling associations with increased inflammation (IL-6, IL-1b and IL-8) that was associated with preterm birth, as well as increased Igs, complement pathway proteins. and mannose binding lectins. Importantly this was also associated with a non-lactobacillus vaginal microbiota (aka bacterial vaginitis). They also assessed the impact of 2 major treatments, either progesterone or cervical cerclage, but didn't find major changes to immunity or complement. This is an extremely important study and may lead to potential targets for decreasing sPTB.

Comments:

1. The authors find changes in Ig subclasses relative to sPTB however it is unclear what these Abs are to. It would greatly benefit the paper to have insights into what these Abs are specific to.

We completely agree that it would be important to establish exactly what the Immunoglobulin classes and subclasses are specific to. We are currently planning a programme of work to determine which vaginal microbiota are being opsonised and by which subclass, and if the degree of opsonisation influences the degree of complement activation/pro-inflammatory cytokine production and the risk of preterm birth. This area of our work is at a very early stage, requires development and optimisation. We anticipate that it will take us at least a year to reach conclusive results. We have acknowledged this as an important area of work to focus on with the following addition to the text in the discussion.

"Future work should focus on determining the degree of reactivity of the Ig classes and subclasses to specific vaginal microbiota to establish if this influences the local immune milieu and ultimately the risk of preterm birth." Lines 303-305.

2. The clustering they used for vaginal communities is not standard, and really not what one would expect in the vaginal microbiome. This is likely because they used V1/V2 instead of V3/V4, which gives more Lactobacillus species identification but less overall genera, and specifically under representation of extremely important and common bacteria such as Gardnerella. If possible, redoing all 16S with V3/V4 is ideal. If this isn't possible this needs to be extremely carefully addressed in the manuscript and more standard vaginal microbiome community groups relative to inflammatory factors and SPTB needs to be re-analyzed.

We agree with the Reviewer that the V1-V2 hypervariable region offers improved resolution of key Lactobacillus species. It is also correct that if a universal forward (27F) primer is used, mismatches can lead to under representation of important vaginal bacterial genera, including Gardnerella. However, as per our study, this can be overcome through the use of a mixed formulation of the 27F forward primer (27F-YM), which has been shown to maintain the rRNA gene ratio of *Lactobacillus* spp. to *Gardnerella* spp(1). We have addressed the rationale for using V1/V2 primer sets in the Methods section.

"This mixed formulation of the 27F forward primer (27F-YM) has been shown to maintain the rRNA gene ratio of Lactobacillus spp. to Gardnerella" see Lines 432-434.

In consideration of the reviewers' comments regarding our clustering strategy, we have now re-classified each vaginal microbiome sample profile using the VALENCIA classifier (VAGinal community state type Nearest Centroid classifier), which is a nearest centroid-based tool that works by classifying samples based on their similarity to a set of reference centroids defined against a set of 13,160 taxonomic profiles. This approach has been validated and tested in multiple different

ethnic populations and is considered a robust method for unbiased, reproducible and standardised reporting of vaginal community state types (CSTs) (2). In addition, we have now reanalysed all reported interactions between CSTs, immune response and preterm birth (Figures 2,3,4,5 6, 7, 9, and Supplementary figures 2,3 and 5). Overall, this reanalysis has confirmed the findings originally reported and in some instances resulted in stronger statistical evidence for interrelationships within the datasets. Dr Samit Kundu, who undertook these analyses, in addition to others within the new version of the manuscript has been added as a co-author. We have updated the Methods section to account for the reviewed analysis. Lines 438-449.

3. Figures 2, 4 and 5 a-f are difficult to interpret. Use of better labeling would be helpful

Thank you for this comment, we have now improved the labelling by increasing the size of the font in the figures, inserting CST I-V labelling, adding titles and dividers, and by altering the colour scheme to be more consistent with the Nature Communications colour scheme.

4. Some of the figure colors are very difficult to see between, and if printed in black and white would be near impossible to see differences, such as green colors in fig 7. But through the authors should address this.

We have now altered the colour scheme in figures 2,3,4,5,9 and Supplementary figures 1,2,3,5 to pastel colours to improve readability if printed in black and white, and to be better aligned to the Nature Communications colour scheme.

Reviewer #2 (Remarks to the Author):

The manuscript contains a lot of data about infection and the immune response in some preterm labor. The assumption that preterm labor is related to infection has been around a while and is generally accepted. That the infection is bacterial in origin has not been shown and the fact that this manuscript does not mention other possible infectious agents makes the data interesting but less important. The figures in general show that almost ever cytokine, immunoglobulin or immune component has a statistical link suggests equally that none of them do or that the statistics used in comparison are not applicable in this instance. The data suggests that the Wilcoxon test may not be appropriate for this data set. The size of the figures and the fact no number are mentioned in the text are off putting.

1. Cytokine, Immunoglobulin and complement concentrations.

The reviewer may have gained the impression that everything that we examined showed significant differences because we presented data relating to analytes which showed significant differences in the main results and relegated those that did not change to the supplementary materials. Importantly, there were many cytokines that were analysed for which no changes in concentrations were seen in women who delivered preterm. These were IFN- γ , TNF- α , GM-CSF, IL-18, IL-4 and IL-5 (Table 2, Supplementary material). We also saw no difference in these cytokine concentrations in relation to microbial composition. We inserted median concentrations and p values comparing samples associated with *Lactobacillus* dominance and depletion in Table 3, supplementary material. In addition, we did not demonstrate any significant changes in IgA concentrations in women who delivered preterm in relation to microbial composition. These results are presented in Supplementary figure 3. We have now also added data on IgE concentrations to Supplementary figure 3. Although IgE concentrations are increased in association with *Lactobacillus* deplete/ CST IV, there is no significant difference in women who deliver preterm. In contrast, we did see significant changes in all of the complement proteins, data of which is presented unaltered in the figures of the main manuscript.

2. Wilcoxon test

The statistical tests used were dependant on the distribution of the data, if the data were paired or not paired, and the number of groups being compared. The Wilcoxon matched pairs signed rank test was used where comparisons were made between two groups and where data followed a

nonparametric distribution and when groups were paired. This was the case when comparing changes in analyte concentrations between timepoints of longitudinal samples (Figure 1 a-d, Figure 4 a-f, Figure 5 a-c, and Figure 9b-i). We have corrected the figure legend in Figure 9 to with the correct statistical test used (Wilcoxon matched pairs signed rank test) rather than Mann-Whitney.

3. Figure sizes and figure number in the text

We have increased the size of font for the labelling to help provide clearer figures, and we have been through the manuscript text and have ensured that each figure is referenced in the text.

4. While the authors suggest that a cross talk is apparent in the study no mechanisms or hypothesis is suggested. The grouping of bacterial agents is compelling but the lack of focus and the number of data points make it difficult to read. A more focused concise report is warranted and an hypothesis driven report warranted

We have now clarified in the Introduction that we hypothesise that the complement system facilitates cross talk between the innate and adaptive immune response to vaginal microbiota.

“We hypothesise that the complement system facilitates cross talk between the innate and adaptive immunity in response to vaginal microbiota“ See Lines 66-67.

Figure 6 represents a linear correlation between MBL and IgM and concentrations of IL-8, IL-6 and IL-1 β , and between C3b/C5 and IL-8, IL-6 and IL-1 β . We feel that this, together with the individual data sets supports our hypothesis. The mechanism to support our hypothesis is summarised in Figure 10 in the form of a schematic which presents our hypothesis that the complement facilitates cross talk between the innate and the adaptive response in microbial driven preterm birth.

We have altered our conclusion to reflect this:

“In conclusion, we have proposed a potential mechanism involving cross talk between the innate and adaptive immune response in microbial driven preterm birth which could lead to the development of novel therapies for the prevention of preterm birth”. Lines 368-370.

Reviewer #3 (Remarks to the Author):

Summary: This manuscript addresses an important topic of high public health relevance. Leveraging a cohort of high-risk pregnant women for spontaneous pre-term birth (sPTB) they showed that maternal host immune response to vaginal microbiota drives inflammation, complement activation, cervical shortening and preterm labor. Specifically, 133 pregnant women with previous sPTB, previous PPRM, previous mid trimester pregnancy loss, previous large loop excision of the transformation zone of the cervix, or a combination of these were prospectively evaluated from early pregnancy (week 16) till delivery. Demographic and clinical profiling including timing of delivery and cervical length by transvaginal ultrasound were performed. Vaginal swabs were obtained for microbiota profile using 16S rRNA sequencing, and for assays of inflammatory cytokines, complement and immunoglobulin analytes. Overall, the study was well designed, and pertinent predictor and outcome variables were captured to address the key research questions. Addressing the following critiques will significantly enhance the manuscript.

1. The control group include participants with risk factors for sPTB who did not experience pre-term delivery – it could be argued that a better control group should be pregnant women with no sPTB risk factor. This control used in the current study could have biased the outcomes and impacted the effect size as they were selected from a population of at-risk individuals for the main outcome of the study.

This study set out to determine if the host immune response to the vaginal microbiota modulated the risk of preterm birth to help address why not all women with a high risk vaginal composition deliver

preterm. Our targeted cohort were women at high risk of preterm birth which represents an important clinical group in whom management is challenging. If novel therapies are to be developed, it is likely that they would be primarily aimed at the women at highest risk of preterm birth initially. An advantage of studying a high-risk group is to be able to enrich the cohort with a high rate of preterm birth (28% compared to the national average of 7%), which helped with achieving statistical power whilst minimising the number of women needed to be recruited. In our experience, women who are at a high risk of preterm birth are more likely to agree to intense sampling protocols (which are in line with the timing of the intense surveillance schedule in our preterm prevention clinic), compared to women at low risk. Nevertheless, it is a very valid point that we agree entirely with, and we have altered the text to explain that a limitation of this study is that caution would be needed in extrapolating the significance of these findings to a low risk population. We have altered the text to read:

“Whilst women who delivered at term without intervention were used as a control group, it is acknowledged that our study cohort was predefined as being at high risk of preterm delivery. Line 91 -93.

“In our study of women at high risk of preterm birth, we demonstrate that this elevation occurs between the early and late second trimester of pregnancy”. Line 237.

2. The study excluded participants with HIV and viral hepatitis but did not screen for nor exclude those with sexually transmitted infections including GC, CT, TV, etc. It is clearly established that STIs could alter the vaginal microbiota and also independently induce inflammatory changes and complement activation in manners that may alter the outcomes measured in the study.

Thank you for this comment which is a valid point. If an abnormal cervical appearance or abnormal discharge was seen on examination (or on clinical history), this would trigger investigation with an STI screen. We did not detect any STIs in this cohort. Based upon UK Public Health England Office of National Statistics data the prevalence of chlamydia and gonorrhoea in our cohort would be expected to be between 0.1%- 0.5% and 0.03-0.1% respectively³, which is in keeping with us not identifying any cases in our study cohort.

We have altered the text to include the following:

“All pregnant women in the UK are screened for syphilis. Otherwise routine screening for sexually transmitted diseases was not included in the study protocol, but would be undertaken in any woman with signs or symptoms.” Lines 381-384.

“No women were diagnosed with a sexually transmitted disease.” Line 96

3. The target population – women of reproductive age group, though pregnant are likely to have diverse vaginal hygiene practices that could affect the composition of the vaginal microbiome – were history of vaginal hygiene captured?

Thank you for this comment. We recommend women not to douche in our preterm prevention clinic when we see them at their first appointment, and we collect data on vaginal hygiene practices in our study populations. We have altered this in the methods section:

“Women were advised to avoid vaginal douching at their first appointment”. Line 384.

4. While the observed differences for some of the outcomes (VMG, cytokines, complements, immunoglobulins) were impressive, the absences of differences in others raised the question of power. Was the study powered a priori for these outcomes? If so, could sample size justification be provided in the statistical section?

We did not power the study a priori. This would have been difficult because this was an exploratory study, and the first to compare local cervicovaginal complement protein and immunoglobulin

concentrations between different vaginal microbial concentrations in pregnancy and in the context of preterm birth.

We accept that there were a few comparisons where there was a change in direction consistent with our hypothesis which did not reach statistical significance. To be in accordance with the journals policy "Where relevant, provide exact values for both significant and non-significant P values" we have now labelled these comparisons with the exact p values, to help the reader to interpret the data . See Figures 3i, 4s,7k and 7s.

"Whilst not all changes reached statistical significance, a trend in accordance with the hypothesis supports clinical significance." Line 190-191

5. A major concern with vaginal cytokine assays using cervicovaginal secretions is the accuracy of the concentration measures. The volume of secretions absorbed by each swab is very likely to be different and may thus influence the estimation of the amount of analytes present – the manuscript in its current form did not address this issue in the methods section.

The supernatant used for immune assays is extracted from the sponge of the swab via a pressurised syringe. The analyte concentrations are reported as picogram or nanogram per ml of fluid absorbed by the swab, and therefore is comparable between patients and timepoints, and is independent of the volume absorbed by the swab. We have updated the methods section for clarity:

"The BBL™ CultureSwab™ was thawed on ice. Supernatant was extracted from the sponge of the swab using a sterile syringe and pressure to release a volume of approximately 350µl. This was centrifuged at 3000g for 10 minutes. Protease inhibitor (5µl/ml; Sigma-Aldrich), was added to the supernatant. This was used for Luminex® immunoassays to quantify the chosen cytokines, complement and immunoglobulin analytes. Results were expressed in picograms or nanograms per ml of cervico-vaginal fluid volume." Lines 409-414.

1. Frank JA, Reich CI, Sharma S, Weisbaum JS, Wilson BA, Olsen GJ. Critical evaluation of two primers commonly used for amplification of bacterial 16S rRNA genes. *Appl Environ Microbiol.* 2008;74(8):2461-70.
2. France MT, Ma B, Gajer P, Brown S, Humphrys MS, Holm JB, et al. VALENCIA: a nearest centroid classification method for vaginal microbial communities based on composition. *Microbiome.* 2020;8(1):166.
3. https://assets.publishing.service.gov.uk/government/uploads/system/uploads/attachment_data/file/914184/STI_NCSP_report_2019.pdf

REVIEWER COMMENTS

Reviewer #1 (Remarks to the Author):

Overall this study is very important. The authors addressed many reviewer concerns, however did so in very "quick and easy" ways, and could've been more thoughtful and thorough in their responses. In particular, there is still concern about the grouping of vaginal microbiome groups and that 4 of 5 groups are Lactobacillus based and not diverse bacteria. They were asked to carefully address this, and adding one line to the text does not justify this. Furthermore, a breakdown of the diverse bacteria should be included at least in supplemental. Given the discrepancies between statistics opinions, the paper should also be reviewed for appropriate statistical analysis.

Overall the figures have been improved, but there is still a lack of referencing appropriate figures in the text, making it difficult to follow the results.

A more thoughtful discussion based on all reviewer comments is warranted.

Reviewer #2 (Remarks to the Author):

Accept as modified.

Reviewer #3 (Remarks to the Author):

The authors have adequately addressed my concerns and the manuscripts is in great shape. i have no additional critique.

REVIEWER COMMENTS

Reviewer #1 (Remarks to the Author):

Overall this study is very important. The authors addressed many reviewer concerns, however did so in very “quick and easy” ways, and could’ve been more thoughtful and thorough in their responses. In particular, there is still concern about the grouping of vaginal microbiome groups and that 4 of 5 groups are *Lactobacillus* based and not diverse bacteria. They were asked to carefully address this, and adding one line to the text does not justify this.

We acknowledge that our previous response may have addressed concerns in insufficient detail. With regards to the grouping of vaginal microbiome groups, we have now used the full VALENCIA classification system of seven CSTs and thirteen sub-CSTs. This now allows for grouping with three *lactobacillus* spp. deplete CSTs and seven deplete sub-CSTs. We have provided a supplementary table which groups all samples in the dataset to the level of the thirteen sub-CSTs, and includes compositional data (Supplementary Table 3). We have also explained the use of the full VALENCIA classification system for our samples in the manuscript in the materials and methods section (lines 494-500), the results section (113-127) and the discussion (279-287). We have re analysed the dataset according to the seven types and thirteen subtypes and have updated the results in the manuscript.

1. No samples were of CST IV-A, CST IV-C0, CST IV-4, and few samples were of IV-C1 and CST IV-C2. We have referred to this in the text on page 127 of the results section, and lines 303-308 of the discussion. This information can be accessed in Supplementary Table 3.
2. We have amended the graph in Figure 2e to represent the percentages of study samples classified in the seven community state types, three groups of which are *Lactobacillus* spp. deplete.
3. We have amended figures 3.e-h, 4.m-r and 5.g-l, and Supplementary figure 3.e-f to include analysis of immune mediators to include CST IV-B and CST IV-C (note CST IV-A absent as no samples of this group in study population).
4. Most of the CST IV samples were either CST IV-B or CST IV-C3, with no CST IV-A, CST IV-C0 or C4, therefore we were unable to perform a sub analysis within the CST IV subtypes. We have acknowledged this in the discussion and highlighted this as an important area of future work. Lines 283-287 and 303-308.
5. We acknowledge the importance of the CST IV subtypes. Despite not having enough numbers to perform a sub analyses, we feel this an important point raised by the reviewer therefore to provide granularity to the analyses of the comparison of immune mediators from women who were CST IV at mid gestation between women who delivered preterm and at term, we coded the datapoints to reflect CST IV-B (n=8), CST IV-C1 (n=1), CST IV-C2 (n=1) and CST IV-C3(n=5). (Figures 3. i-l, 4.s-x, 5.j-l and Supplementary Figure 3.g and h). We also provide the compositional information for these women in Supplementary Table 4).

Furthermore, a breakdown of the diverse bacteria should be included at least in supplemental.

We have provided compositional data, CST and sub-CST for all samples in Supplementary Figure 3. We have also provided a separate table containing compositional data and sub-CST for women who were CST-IV at mid gestation for the analyses of immune mediators between preterm and term (without intervention) in Supplementary Figure 4. In addition, we have supplied a Data Source file with the metadata, the ENA run accession number, composition, CST and SubCST linked by the sample identifier.

Overall the figures have been improved, but there is still a lack of referencing appropriate figures in the text, making it difficult to follow the results.

We have improved the referencing by adding a reference to figures at 17 additional locations in the text and amending 10 for clarity.

A more thoughtful discussion based on all reviewer comments is warranted.

We have added two paragraphs for discussion to address comments. We have added more discussion justifying the use of the V1/V2 over V3/4, see Lines 265 -274.

“In this study we used amplification of the V1/V2 hypervariable region for metataxonomic analysis and clustered the resulting sequence data using the VALENCIA classifier. Amplification of the V1/V2 regions has been widely used in the study of the vagina microbiome, including the original Ravel community state type classification study (1). This approach has the advantage over application of other regions of improved discrimination between species of Lactobacilli, which are the most prevalent genus in the vaginal microbiota. It has the disadvantage of that if a universal forward primer is used, mismatches can lead to under representation of important vaginal bacterial genera including Gardnerella. However, as we have done in this study, this problem can be overcome through the use of a mixed formulation of the 27F forward primer which has been shown to maintain the rRNA gene ratio of Lactobacillus spp. to Gardnerella spp (2).”

We have also added a paragraph in the discussion to address the VALENCIA classification, Lines 275-287, and Lines 303-308.

“Taxonomic profiles of vaginal microbiota communities are commonly sorted into discrete categories termed community state types or vaginal microbiome groups (VMGs) based on the results of hierarchical clustering of the pairwise comparisons. Because the results are dependent on the particular population that was analysed, this approach makes cross study comparisons challenging. The VALENCIA algorithm is a nearest centroid-based tool that works by classifying samples based on their similarity to a set of reference centroids defined against a set of 13,160 taxonomic profiles from 1975 women of reproductive age. This approach has been validated and tested in multiple different ethnic populations and is considered a robust method for unbiased, reproducible, and standardised reporting of vaginal community state types (3). The VALENCIA pipeline characterises individual subject vaginal bacterial communities into seven CSTs of which three represent Lactobacillus spp. deplete communities (CST IV-A, IV-B and IV-C), and thirteen sub-CSTs. Most of our Lactobacillus spp. deplete samples were of IV-B or IV-C3 subtype, with minimal or no samples classed as IV-A, IV-C0-2, and IV-C4. We therefore analysed our data using the seven community state type classification.”

“For the primary analysis of this study, Lactobacillus spp. deplete community states were combined to IV-A, B or C, (with CST IV-C0-4 combined within CST IV-C). Separate analyses were not feasible between CST-IV subtypes due to the limitations in sample number. Whilst our study clearly supports the association between Lactobacillus spp. depletion and inflammation in women delivering preterm, further larger studies will be needed to explore the effects of each of the Lactobacillus spp. deplete subtypes.”

1. Ravel J, Gajer P, Abdo Z, Schneider GM, Koenig SS, McCulle SL, et al. Vaginal microbiome of reproductive-age women. Proc Natl Acad Sci U S A. 2011;108 Suppl 1:4680-7.
2. Frank JA, Reich CI, Sharma S, Weisbaum JS, Wilson BA, Olsen GJ. Critical evaluation of two primers commonly used for amplification of bacterial 16S rRNA genes. Appl Environ Microbiol. 2008;74(8):2461-70.
3. France MT, Ma B, Gajer P, Brown S, Humphrys MS, Holm JB, et al. VALENCIA: a nearest centroid classification method for vaginal microbial communities based on composition. Microbiome. 2020;8(1):166.

REVIEWERS' COMMENTS

Reviewer #5 (Remarks to the Author):

This comprehensive analysis of the vaginal microbiome, immune and complement effectors, adds significantly to our understanding of the interaction between different taxa of the microbiome and women's gynecological and reproductive health. The authors have responded well to the comments of the previous reviews. I see no reason why the manuscript cannot be published as is. My one recommendation might be to move some of the figures (3-7) to supplementary pages. While the figures are informative, they are extremely complicated and probably somewhat superfluous to the text.

Response to reviewer

REVIEWERS' COMMENTS

Reviewer #5 (Remarks to the Author):

This comprehensive analysis of the vaginal microbiome, immune and complement effectors, adds significantly to our understanding of the interaction between different taxa of the microbiome and women's gynecological and reproductive health. The authors have responded well to the comments of the previous reviews. I see no reason why the manuscript cannot be published as is. My one recommendation might be to move some of the figures (3-7) to supplementary pages. While the figures are informative, they are extremely complicated and probably somewhat superfluous to the text.

Author response:

Thank you for your recommendation. We have now moved Figure 4 g-l and Figure 5 d-f to supplementary material. We feel that moving the results showing the influence of *lactobacillus* depletion on microbial mediators of recognition and complement proteins can be placed into the supplementary material to allow more focus and attention to the influence of CST and pregnancy outcome. We feel this now makes the figures less complicated and more focused on the most impactful findings.